# Generative Krylov Subspace Representations for Scalable Quantum Eigensolvers

## Abstract

Predicting ground state energies of quantum many-body systems is one of the central computational challenges in quantum chemistry, physics, and materials science. Krylov subspace methods, such as Krylov Quantum Diagonalization (KQD) and Sample-based Krylov Quantum Diagonalization (SKQD), are promising approaches for this task on near-term quantum computers. However, both require repeated quantum circuit executions for each Krylov subspace and for every new Hamiltonian, posing a major bottleneck under noisy hardware constraints. We introduce Generative Krylov Subspace Representations (GenKSR), a framework that learns a classical generative representation of the entire Krylov diagonalization process. To enable effective modeling of quantum systems, GenKSR leverages a conditional generative model (CGM) framework. We investigate two representative backbone architectures, the standard Transformer and the Mamba state-space model. While Transformers provide a strong baseline, Mamba offers a more computationally efficient alternative due to its linear-time complexity. By learning the distribution of measurement outcomes conditioned on Hamiltonian parameters and evolution time, GenKSR generates Krylov subspace samples for unseen Hamiltonians and for larger subspace dimensions than those used in training. This enables full energy reconstruction purely from the classical model, without additional quantum experiments. We validate our approach through simulations of 15-qubit 1D and 16-qubit 2D Heisenberg models, as well as a hardware experiment on a 20-qubit XXZ chain executed on an IBM quantum processor. Our model successfully learns the distribution from experimental data and generates a high-fidelity representation of the quantum process. This representation enables classical reproduction of experimental outcomes, supports reliable energy estimates for unseen Hamiltonians, and significantly reduces the need for further quantum computation.

## 1 Introduction

The quantum many-body problem is a central challenge across physics (Orús, 2014; Carleo & Troyer, 2017; Gebhart et al., 2023; Torlai & Melko, 2020), chemistry (Barrett et al., 2022; Cao et al., 2019; Kandala et al., 2017; Schütt et al., 2019; Sajjan et al., 2022), and materials science (Carrasquilla, 2020; Takahashi, 2022; Miles et al., 2023). While quantum computers promise to revolutionize computational approaches to these problems, current and near-term devices are limited by noise and imperfections that restrict the size of feasible quantum circuits (Bharti et al., 2022; Sun et al., 2021). Quantum Phase Estimation (QPE) (Nielsen & Chuang, 2010), the canonical quantum eigensolver, offers theoretical guarantees but requires fault-tolerant quantum hardware that is not yet available. This has spurred the development of hybrid quantum-classical algorithms that offload parts of the computation to a classical computer and are better suited to the Noisy Intermediate-Scale Quantum (NISQ) era (Preskill, 2018; Callison & Chancellor, 2022).

The Variational Quantum Eigensolver (VQE) (Peruzzo et al., 2014; McClean et al., 2016) has been a primary candidate, yet it faces significant scalability challenges, including barren plateaus (McClean et al., 2018) in its optimization landscape and substantial measurement overhead for gradient estimation. These limitations motivate the search for alternatives with more reliable convergence behavior.

Krylov Quantum Diagonalization (KQD) (Motta et al., 2020; Stair et al., 2020; Cortes & Gray, 2022; Yoshioka et al., 2025) has emerged as a powerful alternative. In many practical applications, the objective is not to recover the full spectrum but to accurately estimate the ground-state energy (or a few low-lying excited states). Krylov methods exploit this by constructing a compact subspace that retains the information needed for ground-state estimation without resolving the entire spectrum. KQD builds this Krylov subspace via unitary time evolutions on a quantum computer and then classically diagonalizes the Hamiltonian restricted to that subspace. This approach benefits from theoretical convergence guarantees, provided the initial state has sufficient overlap with the true ground state. Furthermore, recent nonasymptotic analyses of quantum circuit noise (Kirby, 2024) proves that the upper bound on the ground-state energy estimation error scales linearly with noise-induced estimation errors, resolving a key misalignment between known numerical results and prior theoretical work (Epperly et al., 2022). While selecting a suitable initial state often relies on physical intuition, systematically improving the accuracy of the energy estimate requires increasing the Krylov dimension $D$. However, this process faces a critical practical bottleneck: each increment of $D$ necessitates a new set of resource-intensive quantum experiments with complex measurement schemes to estimate the projected Hamiltonian $\tilde{H}$ and the overlap matrix $\tilde{S}$. The Sample-based Krylov Quantum Diagonalization (SKQD) (Yu et al., 2025; Piccinelli et al., 2025) partially alleviates these implementation challenges by using the quantum device only to sample bitstrings from Krylov states. The Hamiltonian is then projected onto the corresponding basis subspace entirely in classical post-processing, thereby avoiding the controlled unitaries and Hadamard tests required in KQD and making the method more hardware-efficient.

Nevertheless, a fundamental limitation of instance-specific approaches such as QPE, VQE, KQD, and SKQD is that the entire quantum workflow must be executed anew for each Hamiltonian. This requirement severely limits scalability, as each instance incurs substantial quantum resource costs. In contrast, incorporating machine learning offers a paradigm shift: once a classical model is trained on data generated by a quantum device, it can generalize to unseen Hamiltonians and predict ground-state energy without any further quantum experiments.

Recent advances have demonstrated the potential of generative models for quantum property estimation, including transformer-based approaches (Wang et al., 2022), large language model (Tang et al., 2024), and diffusion-based methods (Tang et al.). These models excel at learning complex probability distributions from measurement data to accurately predict physical properties such as correlations and entanglement entropy. These studies have demonstrated the potential of generative models, but the question of how to integrate them into practical quantum workflows remains open. Beyond generative models, recent Bayesian approaches for VQE (Nicoli et al., 2023) offer an additional learning-based direction.

In this work, we introduce Generative Krylov Subspace Representations (GenKSR), a new paradigm for learning generative representations of quantum Krylov diagonalization (KQD and SKQD) experiments. GenKSR treats the quantum device as a data generator and trains a classical model that captures the conditional distribution of measurement outcomes, enabling quantum property estimation entirely classically. To overcome the scalability limitations of prior Transformer-based approaches with their quadratic ($O(n^2)$) complexity, where $n$ is the number of qubits, GenKSR trains a conditional generative model (CGM) based on the Mamba (Gu & Dao, 2023), which exhibits near-linear ($O(n)$) complexity, on circuit measurement data. Once trained, GenKSR can synthesize realistic subspace samples for unseen Hamiltonians and arbitrary Krylov dimensions, which are then used in the classical KQD and SKQD post-processing pipelines to predict ground-state energy. In this way, GenKSR substantially reduces the need for additional quantum executions.

We validate GenKSR in two settings. First, we perform simulations of KQD on 1D Heisenberg model with up to 15 qubits and SKQD on a two-dimensional $J_1$–$J_2$ Heisenberg model on a $4 \times 4$ square lattice (16 qubits), demonstrating reliable predictions and the ability to extrapolate to larger Krylov dimensions. Second, we conduct SKQD experiments on a 20-qubit XXZ chain using *ibm_fez* quantum processor, showing that the CGM reproduces noisy measurement distributions and achieves energy error statistics comparable to direct experiments under different number of measurement. These results demonstrate that learning generative representations over quantum data enables both generalization across Hamiltonians and substantial reduction in quantum resource cost for many downstream tasks.

Our main contributions are:

- We propose GenKSR, the first framework that unifies KQD and SKQD with generative modeling. By training a classical model on quantum data, GenKSR enables fully classical estimation of the ground-state energies of quantum systems.

- We investigate and benchmark two representative backbone architectures—the standard Transformer and the Mamba state-space model—within the GenKSR framework. We analyze the trade-off between the Transformer's expressive capacity and Mamba's computational efficiency, demonstrating that GenKSR is an architecture-agnostic framework adaptable to different resource constraints.

- We validate GenKSR through simulations of 1D and 2D Heisenberg models up to 16 qubits and SKQD experiments on a 20-qubit IBM quantum processor, demonstrating accurate ground-state energy estimation for unseen Hamiltonians and extrapolation to larger Krylov subspaces than observed in training.

## 2 BACKGROUND

### 2.1 KRYLOV QUANTUM DIAGONALIZATION (KQD)

KQD is a quantum algorithm designed to approximate the ground-state energy of a Hamiltonian $H$ by diagonalizing it within a Krylov subspace of low dimension (Motta et al., 2020; Stair et al., 2020; Cortes & Gray, 2022). The approach generalizes classical Krylov subspace methods, which are widely used in classical computational linear algebra for eigenvalue problems, and adapts them to the quantum setting to extract spectral information from many-body Hamiltonians.

Starting from an initial reference state $|\psi_0\rangle$, the Krylov subspace of dimension $D$ is generated by repeated application of the time evolution operator $U = e^{-iH\Delta t}$ with a fixed timestep $\Delta t$. Defining $|\psi_j\rangle := U^j|\psi_0\rangle$, the resulting Krylov subspace of dimension $D$ is given by $K_D = \mathrm{span}\{|\psi_0\rangle, |\psi_1\rangle, \ldots, |\psi_{D-1}\rangle\}$.

This subspace captures the most relevant components of the dynamics generated by the Hamiltonian and provides a compact basis for approximating ground-state energy. Projecting the Hamiltonian onto this Krylov subspace yields the generalized eigenvalue problem:

$$\tilde{H}v = E\tilde{S}v,$$

where the projected Hamiltonian $\tilde{H}$ and overlap (Gram) matrix $\tilde{S}$ have elements:

$$\tilde{H}_{jk} = \langle\psi_j|H|\psi_k\rangle, \quad \tilde{S}_{jk} = \langle\psi_j|\psi_k\rangle.$$

The smallest eigenvalue $E_0$ of this reduced problem provides an approximation to the ground-state energy of $H$.

The success of KQD, strongly depends on the choice of the initial reference state $|\psi_0\rangle$. In particular, a non-trivial overlap with the true ground state $|\phi_0\rangle$ is essential. If the squared overlap $|\gamma_0|^2 = |\langle\psi_0|\phi_0\rangle|^2$ is too small, theoretical error bounds diverge as $O(1/|\gamma_0|^2)$, leading to unreliable energy estimates (Epperly et al., 2022). On real quantum hardware, weak overlaps exacerbate this issue, as meaningful signals can be overwhelmed by device noise or by numerical instabilities introduced during regularization. In extreme cases, the effective overlap becomes dominated by noise, violating the positivity conditions required for stable energy estimation (Kirby, 2024).

In addition to this overlap dependence, KQD suffers from intrinsic numerical challenges. Because Krylov basis vectors generated by time evolution tend to become nearly linearly dependent (Stair et al., 2020; Epperly et al., 2022), the overlap matrix $\tilde{S}$ approaches singularity. This ill-conditioning renders the generalized eigenvalue problem $\tilde{H}v = E\tilde{S}v$ highly sensitive to statistical noise and Monte Carlo sampling errors, often producing large deviations in the estimated eigenvalues. In fact, classical perturbation theory cannot fully explain the observed behavior of KQD under simultaneous noise and ill-conditioning, underscoring the need for stabilization mechanisms. A widely adopted remedy is eigenvalue thresholding (Epperly et al., 2022; Kirby, 2024; Yoshioka et al., 2025), in which contributions from the eigenspaces of $\tilde{S}$ with eigenvalues below a cutoff $\epsilon$ are removed. This procedure eliminates noise-dominated directions, restores numerical stability, and ensures more reliable eigenvalue recovery. Closely related to canonical orthogonalization, thresholding has been both theoretically justified and empirically validated as an essential component of KQD. Without it, spurious eigenvalues can emerge and corrupt the estimation of the true ground-state energy.

## 2.2 Sample-based Krylov Quantum Diagonalization (SKQD)

KQD has been proposed as a promising variational-free approach for estimating low-energy eigenvalues of quantum many-body Hamiltonians. However, a central limitation of KQD lies in its reliance on explicit estimation of Hamiltonian and overlap matrix elements, $\tilde{H}_{jk} = \langle \psi_j | H | \psi_k \rangle$ and $\tilde{S}_{jk} = \langle \psi_j | \psi_k \rangle$. These estimates typically require Hadamard tests and controlled-unitary operations, which scale poorly with system size and are particularly challenging to implement on noisy processors with limited connectivity. To overcome these obstacles, Sample-based Krylov Quantum Diagonalization (SKQD) (Yu et al., 2025) has been introduced as a hybrid quantum–classical algorithm. SKQD is specifically designed for pre- and early-fault-tolerant quantum computers, aiming to efficiently approximate the ground-state energy by combining the convergence guarantees of KQD with the robustness of sample-based quantum diagonalization (SQD) (Kanno et al., 2023; Robledo-Moreno et al., 2025; Barison et al., 2025; Danilov et al., 2025) methods. By relying on projective measurement outcomes rather than matrix-element estimation, SKQD achieves substantially reduced circuit depth and improved resilience against hardware noise. The SKQD workflow proceeds as follows:

1. For each Krylov dimension $k \in \{0, \ldots, D - 1\}$, prepare the state $|\psi_k\rangle = U^k |\psi_0\rangle$ on the quantum computer.

2. For each state $|\psi_k\rangle$, collect a set of $M$ measurement outcomes (bitstrings) $\{a_{km}\}_{m=1}^M$ by measuring in the computational basis.

3. Construct a combined subspace $B$ spanned by the union of all unique bitstrings sampled across all Krylov dimensions: $B = \text{span}\{|a_{km}\rangle\}$.

4. Classically project the Hamiltonian $H$ onto the subspace $B$ to obtain $H_{\text{proj}} = P_B H P_B$, where $P_B$ is the projector onto $B$.

5. Solve the reduced eigenvalue problem by classical diagonalization of $H_{\text{proj}}$, extracting the lowest eigenvalue as the ground state energy estimate.

The theoretical foundation of SKQD rests on two well-established assumptions. First, SKQD inherits the convergence guarantees of Krylov methods (Yu et al., 2025; Piccinelli et al., 2025): if the initial state $|\psi_0\rangle$ has a polynomial overlap with the true ground state $|\phi_0\rangle$, then the ground-state energy can be approximated within polynomial time. Second, its efficiency relies on the sparsity of the ground state in the computational basis, meaning that the probability mass of the wavefunction is concentrated on a polynomially sized subset of the exponentially large Hilbert space. Under these conditions, the most significant basis states can be efficiently captured through sampling, ensuring that the classically constructed subspace $B$ provides a faithful representation for diagonalization. This principle has already been successfully applied in quantum-centric supercomputing architectures to address large-scale quantum chemistry problems beyond the reach of exact diagonalization (Robledo-Moreno et al., 2025).

## 3 Method

### 3.1 Generative Krylov Subspace Representation

The main limitation of both KQD and SKQD is that quantum circuits must be executed separately for each Krylov dimension $D = 1, 2, \ldots, D_{\max}$ and for every new Hamiltonian, requiring additional unitary time evolutions and extensive measurements, which results in substantial overhead on near-term quantum device. Our goal is to replace this iterative quantum workflow with a classical generative model that captures the underlying measurement distribution and can efficiently generate samples for arbitrary Hamiltonians and Krylov dimensions.

Let $\mathbf{x}$ denote Hamiltonian parameters (e.g., coupling strengths of a spin model) and $t_l = l \cdot \Delta t$ represent a time evolution step. We aim to approximate the conditional probability distribution of measurement outcomes $\vec{a} = (a_1, \ldots, a_n)$ as $p_\theta(\vec{a} \mid \mathbf{x}, t_l) \approx p(\vec{a} \mid \mathbf{x}, t_l)$, where $p(\vec{a} \mid \mathbf{x}, t_l)$ denotes the true quantum measurement distribution and $p_\theta$ is its generative model parameterized by neural network weights $\theta$. Once trained, this model can generate measurement samples for unseen Hamiltonians or extended Krylov dimensions, enabling full reconstruction of the ground-state energy entirely classically, without additional quantum executions (see Figure 1).

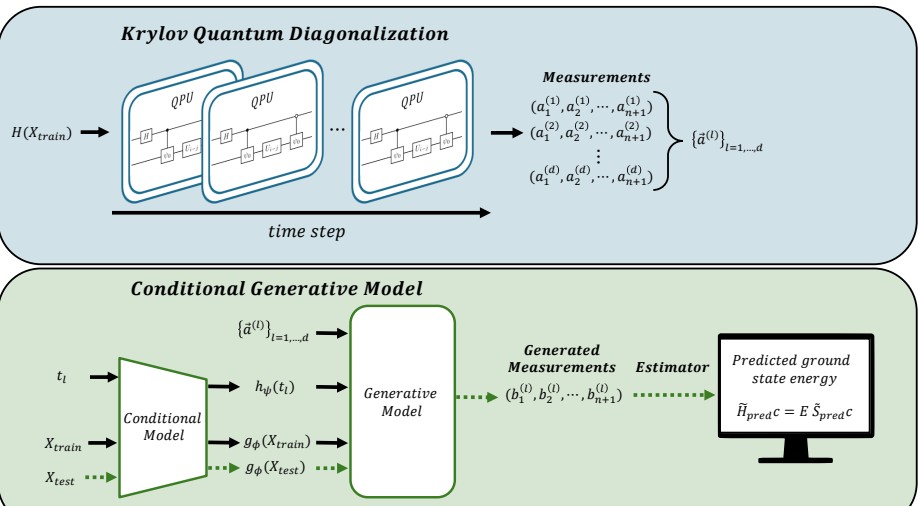

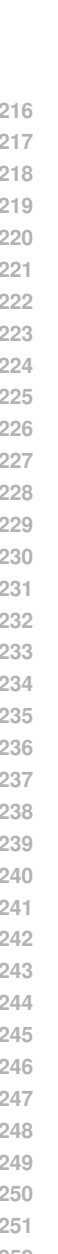

Figure 1: The workflow of GenKSR. The framework consists of a data generation step on a quantum computer (top panel, blue) and a generative modeling step on a classical computer (bottom panel, green). **Training (solid black arrows):** Krylov subspace for a set of training Hamiltonian $X_{\text{train}}$ are prepared and measured on a QPU to collect measurement outputs $\vec{a}^{(l)}$, which are used to train a conditional generative model. **Inference (dashed green arrows):** For a new Hamiltonian $X_{\text{test}}$, the trained model generates synthetic measurement samples for a chosen evolution time $t_l$, which are then processed by a classical estimator to reconstruct the ground-state energy.

This approach, which we refer to as **Generative Krylov Subspace Representation (GenKSR)**, provides a representation of the Krylov subspace process.

## 3.2 MODEL ARCHITECTURE

To model $p_\theta(\vec{a} \mid \mathbf{x}, t_l)$, we adopt a conditional autoregressive modeling framework, following the approach of CGM for quantum states (Wang et al., 2022). The joint probability distribution over measurement outcomes is factorized as $p_\theta(\vec{a} \mid \mathbf{x}, t_l) = \prod_{i=1}^{n} p_\theta(a_i \mid a_1, \ldots, a_{i-1}, \mathbf{x}, t_l)$. Each conditional factor is modeled by a conditional neural network. Building on earlier CGM approaches that use Transformer architectures, we evaluate both Transformer and Mamba as alternative backbone models for GenKSR. Figure 2 illustrates the Mamba-based variant; the Transformer version is identical except that the Mamba block is replaced by a self-attention block.

**Input and positional embeddings.** Partial measurement outcomes $(a_1, \ldots, a_{i-1})$ are embedded into dense vectors via a token embedding layer. Standard sinusoidal positional encodings are added to encode the position $i$ in the sequence.

**Conditional Embedder.** We incorporate Hamiltonian parameters and time evolution steps as conditioning variables.

- **Hamiltonian embedding** $g_\phi(\mathbf{x})$**:** For Hamiltonians that admit a graph representation— such as those with two-body interactions—we construct an interaction graph and use a graph convolutional network (GCN) (Kipf, 2016) to extract a graph-level embedding. This, in principle, allows the conditional network to capture variations in geometry, coupling structure, and two-body Pauli interaction types within a unified encoding scheme.

- **Time embedding** $h_\psi(t_l)$**:** The discrete evolution index $t_l$ is mapped into a dense vector via a linear layer.

The two embeddings are combined to form a global context vector $\mathbf{c} = g_\phi(\mathbf{x}) + h_\psi(t_l)$.

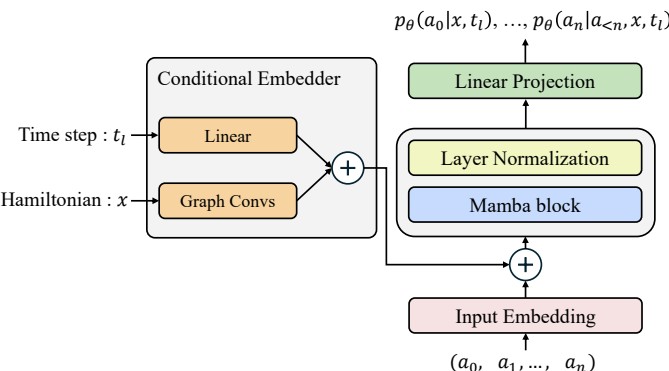

$$p_\theta(a_0|x,t_l), ..., p_\theta(a_n|a_{<n}, x, t_l)$$

Figure 2: Overview of the Mamba based CGM

**Fusion with token sequence.** The context vector $\mathbf{c}$ is added to each input token embedding, ensuring that the generation process at every qubit position is conditioned on both the Hamiltonian and the Krylov step.

**Mamba blocks.** The conditioned sequence is processed by a stack of $N$ Mamba blocks. Unlike the quadratic-cost self-attention in Transformers (Vaswani et al., 2017), Mamba relies on structured state-space models (SSMs) that operate in linear time, making the model scalable to larger qubit numbers while maintaining modeling capacity.

**Output Layer.** Finally, the output vector from the last Mamba block corresponding to the $i$-th position is passed through a linear layer followed by a softmax function. This produces a valid probability distribution over the possible outcomes for the $i$-th qubit, from which a sample can be drawn.

### 3.3 TRAINING AND INFERENCE

Given a dataset $\mathcal{D} = \{(\vec{a}, \mathbf{x}, t_l)\}$ of measurement outcomes, Hamiltonians, and time evolution step, the model is optimized by minimizing the negative log-likelihood:

$$\mathcal{L} := -\mathbb{E}_{(\mathbf{x}, t_l, \vec{a}) \sim \mathcal{D}} \left[ \log p_{\theta, \phi, \psi}(\vec{a} \mid \mathbf{x}, t_l) \right].$$

The training dataset $\mathcal{D}$ consists of measurement outcomes collected from simulated or hardware-executed quantum circuits across multiple Hamiltonians and time evolution step. For each experiment, we sample bitstrings either in the computational basis or using a Pauli-6 positive operator-valued measure (POVM) (Huang et al., 2020) depending on the setup, and pair them with the corresponding Hamiltonian parameters and time evolution step.

Once trained, inference proceeds entirely classically. Given a new Hamiltonian $\mathbf{x}^*$ and desired Krylov dimension, the model generates measurement samples $\{\vec{a}^{(l)}\}$ consistent with the learned distribution $p(\vec{a} \mid \mathbf{x}^*, t_\ell)$. These generated samples can be directly integrated into downstream KQD or SKQD post-processing pipelines. Specifically, in SKQD we project the Hamiltonian onto the span of generated bitstrings, while in KQD the generated samples are processed using classical shadow estimators (Huang et al., 2020) to reconstruct the expectation values corresponding to the elements of the projected Hamiltonian and overlap matrix. Crucially, this allows us to estimate ground state energies for unseen Hamiltonians and extended Krylov dimensions without executing additional quantum circuits.

This training–inference paradigm provides two primary advantages. First, it reduces the quantum resource requirements by replacing repeated circuit executions with classical generation. Second, it adapts to hardware noise by learning directly from experimental distributions, ensuring that inference faithfully reflects realistic device behavior. Together, these properties establish GenKSR as a practical and scalable framework for accelerating Krylov-based quantum diagonalization.

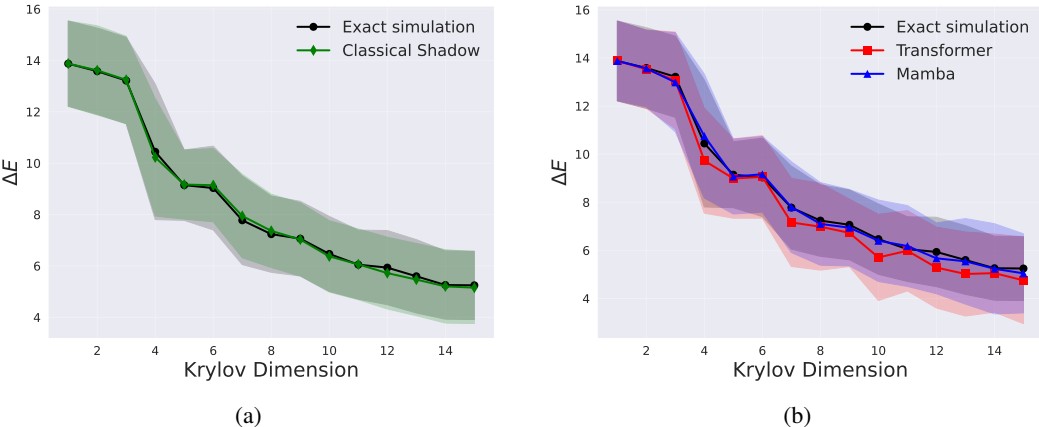

Figure 3: KQD energy prediction for a 15-qubit Heisenberg model, averaged over 20 unseen test Hamiltonians. **(a)** Comparison of the CS with the exact simulation. **(b)** Comparison of the trained Transformer and Mamba models with the exact simulation. Models were trained on data for $D \leq 5$ and evaluated up to $D = 15$ using 10,000 measurement shots..

## 4    EXPERIMENTS

We validate our GenKSR framework in two distinct settings. First, we employ noiseless classical simulations to benchmark the model's generalization and extrapolation capabilities, applying KQD to 1D and SKQD to 2D Heisenberg models. Second, we use a real quantum processor to demonstrate that GenKSR can learn a faithful representation of a SKQD experiment, showcasing its practical utility.

### 4.1    SIMULATION: KQD ON HEISENBERG MODEL

We first validate our approach using simulated data, where we have access to the exact ground truth. We consider the 1D anti-ferromagnetic Heisenberg model with 5, 10, and 15 qubits. The Hamiltonian is given by:

$$H(x) = \sum_{\langle i,j \rangle \in E} x_{ij}(X_i X_j + Y_i Y_j + Z_i Z_j),$$

where the weights $x_{ij}$ are independently sampled from the uniform distribution $U[0,2]$. We generated a dataset of 50 such Hamiltonians, using 30 for training and 20 for testing. For each Hamiltonian, we performed exact diagonalization to obtain the ground state energy and simulated the KQD algorithm up to $D = 15$, generating measurement outputs from the Pauli-6 POVM.

**Training and Extrapolation Protocol.**    We employed a consistent training protocol for all generative models (see Appendix A for full experimental details). Each model was trained using measurement data from the first five Krylov dimensions $D \in \{1, \ldots, 5\}$. We then evaluated the models on 20 unseen test Hamiltonians by generating measurement samples up to $D = 15$. This setup is designed to test two key capabilities: generalization to unseen Hamiltonians and extrapolation to larger Krylov subspace beyond those seen during training. The entire procedure was repeated for 1,000, 5,000, and 10,000 measurement shots per Krylov subspace to test the models' sensitivity to different numbers of samples. For instance, in the 10,000-shots case, each model was trained on a total of $30(\text{Hamiltonians}) \times 5(\text{Krylov dimensions}) \times 10,000(\text{shots}) = 1,500,000$ measurement samples.

**Results and Analysis.**    Figure 3 illustrates the qualitative performance on a representative 15-qubit test Hamiltonian, using models trained on 10,000 measurement shots. The accuracy of the estimates is quantified using the absolute ground-state energy error, defined as $\Delta E = |\hat{E}_0 - E_0|$, where $\hat{E}_0$ is the predicted ground state energy from each method and $E_0$ is the true (exact) ground state energy

Table 1: RMSE comparison for KQD energy prediction across different qubit numbers and measurement shots.

| | 5 qubit | | | 10 qubit | | | 15 qubit | | |
|---|---|---|---|---|---|---|---|---|---|
| Model | Shots = 1,000 | 5,000 | 10,000 | 1,000 | 5,000 | 10,000 | 1,000 | 5,000 | 10,000 |
| Transformer | 0.634 | 0.229 | 0.234 | 1.729 | 1.217 | 1.686 | 2.475 | 1.415 | 1.060 |
| Mamba | 0.435 | 0.222 | 0.225 | 1.634 | 0.958 | 0.771 | 2.849 | 1.432 | 0.924 |
| Classical Shadow | 0.434 | 0.084 | 0.027 | 1.858 | 0.884 | 0.451 | 2.717 | 0.991 | 0.671 |

of the Hamiltonian. Figure 3a compares the energy curve obtained from the Classical Shadow (CS) baseline against the exact simulation. Both are derived from the same underlying quantum circuits; the exact simulation evaluates expectation values analytically without sampling noise, whereas CS uses randomized measurements with finite shots, thereby reflecting statistical and reconstruction errors. As a baseline, CS can approach the exact simulation results when a sufficient number of measurement shots is used (see Appendix B).

Figure 3b shows the predictions from our trained Transformer and Mamba models. Both generative models accurately reproduce the energy curve within the training regime (($D \leq 5$) and successfully extrapolate the convergence behavior for higher, unseen dimensions ($D > 5$). Notably, the Mamba-based model tracks the exact simulation more closely than the Transformer, indicating a superior ability to capture the underlying quantum dynamics and generalize beyond the training regime.

To provide a quantitative comparison, we use the Root Mean Square Error (RMSE), aggregated across all test samples and Krylov dimensions. For each of the $N = 20$ test Hamiltonians ($s = 1, \ldots, N$) and each Krylov dimension $d \in \{1, \ldots, 15\}$, we compare the energy predicted by a given method, $E_{s,d}^{(\text{method})}$, to the exact simulation energy, $E_{s,d}^{(\text{exact sim})}$. The overall RMSE is then calculated as RMSE $= \sqrt{\frac{1}{ND} \sum_{s=1}^{N} \sum_{d=1}^{D} \left( E_{s,d}^{(\text{exact sim})} - E_{s,d}^{(\text{method})} \right)^2}$.

Table 1 summarizes the results. Overall, the Mamba-based model tends to achieve lower RMSE values compared to the Transformer, particularly as the number of measurement shots increases. While the CS baseline remains strong, the Mamba model demonstrates competitive performance and highlights the potential of learning a generative representation.

### 4.2 SIMULATION: SKQD ON THE TWO-DIMENSIONAL $J_1$–$J_2$ HEISENBERG MODEL

To further evaluate the generalization capability of GenKSR beyond one-dimensional systems, we performed SKQD simulations on the two-dimensional $J_1$–$J_2$ Heisenberg model on a $4 \times 4$ square lattice (16 qubits) with periodic boundary conditions. The Hamiltonian is defined as

$$\hat{H} = J_1 \sum_{\langle r,r' \rangle} \hat{\mathbf{S}}_r \cdot \hat{\mathbf{S}}_{r'} + J_2 \sum_{\langle\langle r,r' \rangle\rangle} \hat{\mathbf{S}}_r \cdot \hat{\mathbf{S}}_{r'},$$

where $\langle r, r' \rangle$ and $\langle\langle r, r' \rangle\rangle$ denote nearest- and next-nearest-neighbor pairs, respectively. We fixed $J_1 = 1$ and sampled $J_2$ uniformly from $[0, 1]$. We generated a dataset of 50 such Hamiltonians, using 30 for training and 20 for testing. For each Hamiltonian and Krylov step, we collected 1,000 measurement samples, using data up to Krylov dimension $D \leq 15$ for training.

After training, we evaluated the ability of GenKSR to extrapolate beyond the training regime by generating samples for Krylov dimensions up to $D = 30$ for all test Hamiltonians and performing SKQD post-processing. A comparison of the distribution of energy errors at fixed Krylov dimensions is provided in Appendix A.4, where GenKSR is shown to closely reproduce the SKQD baseline.

To further benchmark GenKSR against established variational approaches, we evaluated the Heisenberg model with a fixed parameter $J_2 = 0.5$. In this setting, we compared three variants of our framework—(i) exact SKQD simulation, (ii) Transformer-based GenKSR, and (iii) Mamba-based GenKSR—against a variational neural network quantum state based on a Vision Transformer (ViT-NQS) Rende et al. (2024).

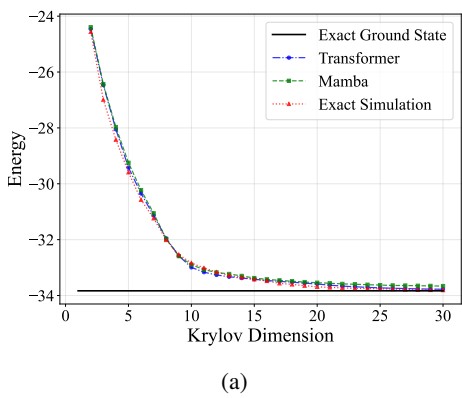
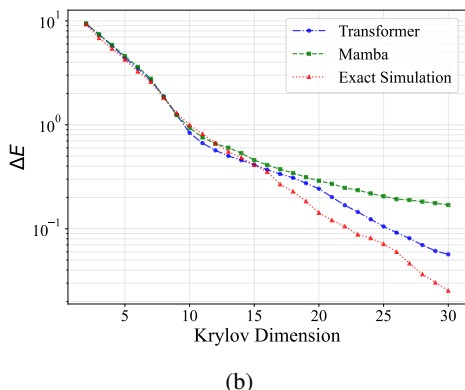

(a)                                 (b)

Figure 4: Performance of exact SKQD, Transformer GenKSR, and Mamba GenKSR on the $4 \times 4$ $J_1$–$J_2$ model at $J_2 = 0.5$. (a) Ground-state energy versus Krylov dimension $D$. (b) Energy error $\Delta E$ in semi-logarithmic scale.

Figure 4 shows the energy estimates. As shown in Figure 4a, the exact SKQD baseline exhibits exponential convergence of the energy with increasing Krylov dimension $D$. Both Transformer and Mamba accurately track this convergence trend, closely following the energy decay of the exact simulation. Figure 4b shows the energy error $\Delta E$ on a logarithmic scale, highlighting that Transformer reproduces the exponential decay of SKQD, while Mamba GenKSR shows a slightly slower decay when extrapolating to larger $D$.

To benchmark against variational approaches, we trained a ViT-NQS using variational Monte Carlo (VMC) on the same Hamiltonian. The training curve is provided in Appendix A.4, and the final converged energy is compared with GenKSR and the exact SKQD baseline in Table 2.

Table 2: Comparison of ground-state energy per site for the $4 \times 4$ $J_1$–$J_2$ model at $J_2 = 0.5$.

|  | ViT-NQS | Mamba (GenKSR) | Transformer (GenKSR) | Exact SKQD |
|---|---|---|---|---|
| Energy per site | $-0.5136$ | $-0.526$ | $-0.5277$ | $-0.5282$ |

These results highlight two key points. First, Transformer GenKSR generalizes beyond its training dimensions, accurately capturing the exponential convergence of SKQD even in frustrated 2D systems. Second, both GenKSR variants outperform the ViT-NQS, with Transformer GenKSR producing results nearly identical to exact SKQD.

### 4.3   Hardware Experiment: SKQD on a 20-Qubit XXZ Chain

To demonstrate the viability of our approach on real quantum hardware, we performed SKQD experiments for a 20-qubit anti-ferromagnetic XXZ chain with periodic boundary conditions on the *ibm_fez* quantum processor (see Appendix A.5 for hardware details). The Hamiltonian is defined as:

$$H = \sum_{i,j}^{N} J_{xy} \left( X_i X_j + Y_i Y_j \right) + Z_i Z_j,$$

where the coupling term $J_{xy}$ was sampled uniformly at random from the interval $[0, 1]$. We generated a dataset of 100 such Hamiltonians, using 70 for training and 30 for testing.

For each of the 70 training Hamiltonians, we executed experiments for Krylov dimensions $D = 1$ through $D = 5$, collecting 1,000 measurement shots at each dimension. These measurement bitstrings were then used to train both a Transformer-based CGM and a Mamba-based CGM. At evaluation, we compared the energy error distributions obtained from direct hardware measurements against those reconstructed by the two generative models for the unseen 30 test Hamiltonians.

As shown in Figure 5, both generative models reproduce the error distributions across different number of samples (1k, 5k, and 10k samples). Importantly, this agreement holds not only for the

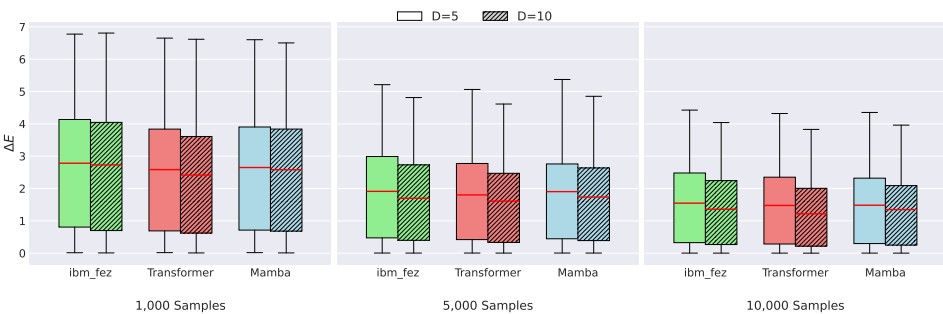

Figure 5: Comparison of energy error distributions on the 20-qubit *ibm_fez* processor. The distributions are shown for samples obtained directly from the *ibm_fez*, and those generated by Transformer and Mamba. Each panel corresponds to a different number of measurement samples (1k, 5k, 10k). Within each panel, the color of each box plot indicates the sample source, while the hatch pattern distinguishes the Krylov dimension: D=5 (no hatch) versus the extrapolated D=10 (hatched). Each box plot displays the median (red line), interquartile range, and data range over 30 test Hamiltonians.

training regime at $D = 5$ but also for the extrapolated setting at $D = 10$. These results demonstrate that our conditional generative modeling approach can capture the noisy statistics of real hardware measurements and extend them to larger Krylov dimensions, thereby reducing the need for repeated quantum runs.

While both models achieve comparable accuracy on this 20-qubit system, a critical consideration for future applications is computational scalability. The Mamba architecture, with its near-linear time complexity ($O(n)$), presents a significant theoretical advantage over the Transformer's quadratic complexity ($O(n^2)$) for modeling larger quantum systems (see Appendix A.3). Therefore, our results not only validate the GenKSR framework on real hardware but also highlight the Mamba architecture as a highly promising and scalable path forward for building classical surrogates of large-scale quantum computations.

## 5 DISCUSSION AND CONCLUSION

In this work, we introduced GenKSR, a framework that integrates generative modeling with quantum Krylov diagonalization. We investigated two representative backbone architectures, the Transformer and the Mamba. Our results highlight a trade-off between performance and computational cost. The Transformer demonstrated superior accuracy in capturing complex correlations in two-dimensional systems, whereas Mamba provided a more computationally efficient path to larger systems through its linear-time complexity. We note that GenKSR is architecture-agnostic and compatible with other efficient backbones like linear-attention Transformers (Katharopoulos et al., 2020; Choromanski et al., 2020). Through both simulation and hardware experiments, we demonstrated that GenKSR not only generalizes to unseen Hamiltonians but also extrapolates to higher Krylov dimensions while faithfully reproducing noisy experimental data. Taken together, these results position GenKSR as a scalable framework that accelerates Krylov-based eigensolvers by shifting the computational burden: the quantum computer acts primarily as a data generator, while inference and energy estimation are offloaded entirely to classical hardware, thereby significantly reducing quantum resource costs.

Several promising directions remain for future work. Integrating GenKSR with quantum error mitigation and suppression techniques (Temme et al., 2017; Kim et al., 2020; 2023; Lee & Park, 2023; Liao et al., 2024) could enable the model to learn representations that more closely reflect noise-reduced behavior. Further improvements could come from systematic ablation studies to better understand the role of conditioning mechanisms and model hyperparameters. Incorporating physics-informed training objectives, such as variational energy minimization, may also help steer the generative model toward low-energy subspaces. Another interesting direction is exploring whether broader training or more expressive conditional networks can further strengthen the generalization capabilities of GenKSR across different Hamiltonian families. Finally, extending GenKSR to excited-state estimation and non-equilibrium quantum dynamics would broaden its applicability across quantum simulation tasks.

## REPRODUCIBILITY STATEMENT

We have provided detailed descriptions of the algorithms, datasets, and experimental setups to ensure reproducibility. The datasets generated and analyzed during the current study are available from the corresponding author on reasonable request. The implementation code and scripts for reproducing the results will be released upon publication.

## USE OF LARGE LANGUAGE MODELS

We used large language models to improve the clarity and grammar of the text and make minor code edits.

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

## A  EXPERIMENTAL DETAILS

### A.1  QUANTUM CIRCUITS FOR KRYLOV STATE PREPARATION AND MEASUREMENT

This appendix details the structure and derivation of the quantum circuits used to implement the KQD algorithm. The circuits are designed to efficiently prepare the Krylov basis states $|\psi_k\rangle$ and measure the necessary matrix elements for constructing the projected Hamiltonian $\tilde{H}$ and overlap matrix $\tilde{S}$. The methodology is based on the efficient implementation described in the supplementary materials of  Yoshioka et al. (2025).

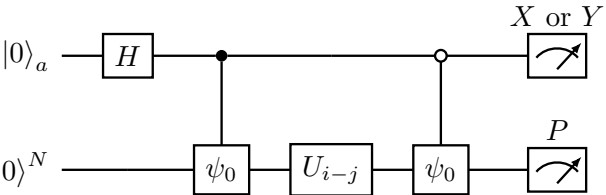

Figure 6: Efficient Hadamard test circuit using an auxiliary qubit to extract the real ($X$) and imaginary ($Y$) parts of matrix elements.

The core task of the quantum processor is to estimate the matrix elements $\tilde{H}_{ji} = \langle\psi_j|H|\psi_i\rangle$ and $\tilde{S}_{ji} = \langle\psi_j|\psi_i\rangle$. These can be rewritten using the time evolution operator $U_k = e^{-ikH\Delta t}$ and the initial state $|\psi_0\rangle$ as:

$$\tilde{S}_{ji} = \langle\psi_0|U_j^\dagger U_i|\psi_0\rangle = \langle\psi_0|U_{i-j}|\psi_0\rangle,$$

$$\tilde{H}_{ji} = \langle\psi_0|U_j^\dagger H U_i|\psi_0\rangle = \langle\psi_0|H U_{i-j}|\psi_0\rangle.$$

Here, we have used the property $U_j^\dagger = e^{ijH\Delta t}$ and the fact that $H$ commutes with the time-evolution operator. Both matrix elements can be expressed as expectation values of the form $\langle\psi_0|PU_{i-j}|\psi_0\rangle$, where $P$ is either the identity operator $I$ (for $\tilde{S}$) or a Pauli operator from the Hamiltonian decomposition $H = \sum_p \alpha_p P_p$ (for $\tilde{H}$).

To measure these quantities, a Efficient Hadamard test is employed, as illustrated in Figure 6. The circuit utilizes an auxiliary qubit to measure the real and imaginary parts of the desired matrix element. The state of the system just before measurement is:

$$\frac{1}{\sqrt{2}}\left(e^{i\phi}|0\rangle_a|\psi_0\rangle + |1\rangle_a U_{i-j}|\psi_0\rangle\right),$$

where the phase $\phi$ arises from the action of $U_{i-j}$ on the $|0\rangle^N$ state and is classically calculable. By measuring the auxiliary qubit in the $X$ or $Y$ basis, we can extract the desired expectation values:

$$\langle X_a \otimes P\rangle = \mathrm{Re}[e^{-i\phi}\langle\psi_0|PU_{i-j}|\psi_0\rangle],$$

$$\langle Y_a \otimes P\rangle = \mathrm{Im}[e^{-i\phi}\langle\psi_0|PU_{i-j}|\psi_0\rangle].$$

Since $\phi$ is known, we can reconstruct the full complex value of the matrix element from these measurement outcomes.

### A.2  IMPLEMENTATION PARAMETERS

**Initial State Preparation.**  For the implementation of this circuit, specific choices were made for the initial state preparation and the time evolution. The initial state $|\psi_0\rangle$ for all experiments was the Néel state, an antiferromagnetic state with an alternating spin configuration, i.e., $|\psi_0\rangle = |1010\ldots\rangle$. This state is a computational basis state and is efficiently prepared using single-qubit $X$ gates.

**Time Evolution.**  The time evolution operator $U_{i-j}$ was approximated using a Trotter-Suzuki decomposition, as the terms in the Hamiltonian do not generally commute. To ensure a balance between accuracy and circuit depth, a second-order Trotter formula was used, and the number of Trotter steps was fixed to 6 for all experiments.

The selection of the time evolution step, $\Delta t$, is a critical parameter that involves a trade-off between algorithmic accuracy and numerical stability. As shown in Epperly et al. (2022), a sufficiently small timestep is necessary to prevent contributions from high-energy states from corrupting the Krylov subspace. This analysis establishes $\Delta t \approx \pi/\|H\|$ as a reliable guideline and suggests that it is preferable to underestimate this value rather than overestimate it.

However, choosing an excessively small $\Delta t$ is also detrimental, as it leads to a poorly conditioned Krylov subspace. When the time evolution step is too short, the resulting basis vectors become nearly linearly dependent, which can destabilize the process of solving the generalized eigenvalue problem. Balancing these competing factors, we set the timestep for all our experiments to $\Delta t = \pi/\|H\|$.

### A.3 COMPARATIVE ANALYSIS OF COMPUTATIONAL COST

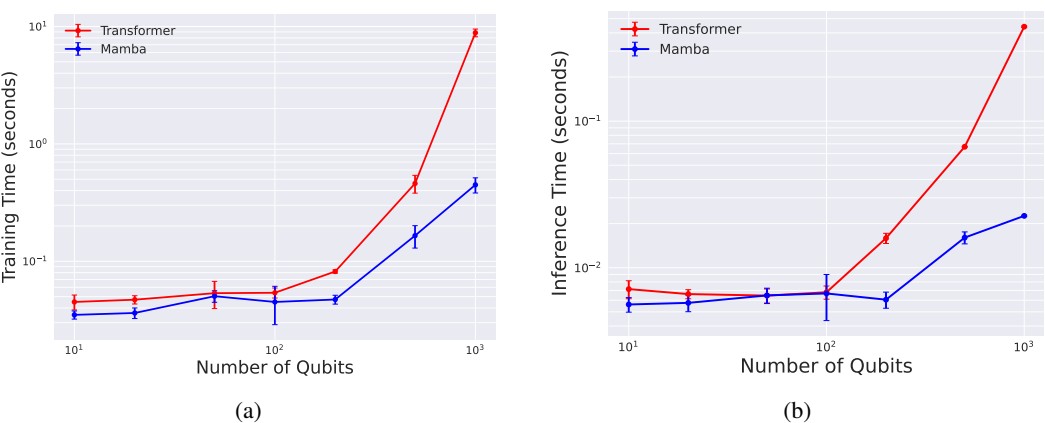

(a)                                                            (b)

Figure 7: Training and inference time as the number of qubits increases. (a) Training time (in seconds) and (b) inference time (in seconds) are shown for the Transformer (red) and Mamba (blue) models.

### A.4 TWO-DIMENSIONAL $J_1 - J_2$ HEISENBERG MODEL RESULTS

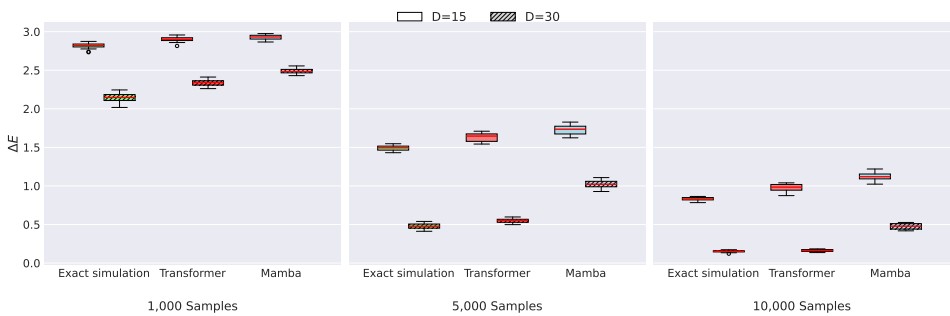

Figure 8: Comparison of energy error distributions on the two-dimensional $J_1 - J_2$ Heisenberg model on a $4 \times 4$ square lattice. The distributions are shown for exact SKQD simulations, and for samples generated by Transformer and Mamba. Each panel corresponds to a different number of measurement samples (1k, 5k, 10k). Within each panel, the color of each box plot indicates the sample source, while the hatch pattern distinguishes the Krylov dimension: D=15 (no hatch) versus the extrapolated D=30 (hatched). Each box plot displays the median (red line), interquartile range, and data range over 20 test Hamiltonians.

Figure 8 summarizes the ground-state energy errors, comparing exact SKQD, Transformer-based GenKSR, and Mamba-based GenKSR. The results show that both generative models successfully capture the energy distributions of the 2D lattices. Transformer GenKSR in particular reproduces

the SKQD baseline with high accuracy, even when extrapolated to Krylov dimensions twice those seen during training. These findings demonstrate that GenKSR generalizes effectively to higher-dimensional systems and can model the richer correlation structures present in 2D lattices.

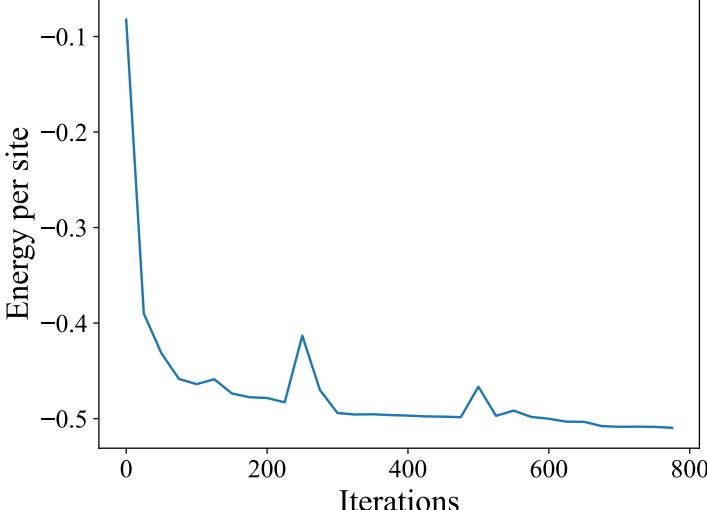

Figure 9: Energy per site as a function of VMC iterations during training of the ViT-NQS model on the two-dimensional $J_1$−$J_2$ Heisenberg model on a $4 \times 4$ square lattice at $J_2 = 0.5$.

### A.5 EXPERIMENTAL HARDWARE

All hardware experiments were executed on the *ibm_fez* quantum processor, a 156-qubit superconducting device with a heavy-hexagon lattice topology. To select a suitable subset of qubits, we employed the Qiskit transpiler with `optimization_level=3`, which identifies an optimal 20-qubit subgraph given the circuit structure and device connectivity (Javadi-Abhari et al., 2024). The physical arrangement of this chosen subset is shown in Figure 10.

We characterized the baseline performance of this subset using calibration data available at the time of execution. The qubits exhibited a median $T_1$ relaxation time of $140 \, \mu$s and a median $T_2$ coherence time of $101 \, \mu$s. The readout assignment error was approximately $1.3\%$ with a measurement duration of $2 \, \mu$s. Single-qubit `sx` gates showed a median error rate of $0.025\%$ with a gate length of $24 \, $ns, while two-qubit `cz` gates, which are most relevant for our workloads, had a median error of $0.25\%$ and a gate duration of $68 \, $ns.

## B  THEORETICAL ANALYSIS USING CLASSICAL SHADOW

### B.1 CONSTRUCTING CLASSICAL SHADOWS

We analyze the scalability of our approach using the framework of classical shadow (Huang et al., 2020) to characterize the output state of the KQD algorithm for each Krylov dimension, $k \in \{0, 1, \ldots, d - 1\}$. Let the $n$-qubit state obtained at a given dimension $k$ be denoted by $\rho_k$. We construct its classical shadow, $\mathcal{S}_{\rho_k}$, by repeating a randomized measurement procedure $M$ times on identical copies of $\rho_k$. Each of the $M$ measurement trials proceeds as follows.

1. **Random Pauli measurements.** For each qubit, uniformly select a measurement basis from $\{X, Y, Z\}$ and perform the corresponding projective measurement on $\rho_k$.

2. **Classical Snapshot:** Each trial yields a bitstring outcome $\hat{b} = (\hat{b}_1, \ldots, \hat{b}_n)$. From this data we form a classical snapshot

$$\hat{\rho}_k = \mathcal{M}^{-1}(U^\dagger |\hat{b}\rangle\langle\hat{b}| U),$$

Figure 10: Layout of the 20-qubit subset selected from the *ibm_fez* processor.

where $U = U_1 \otimes \cdots \otimes U_n$. Crucially, this snapshot is an unbiased estimator of the true state, i.e., $\mathbb{E}[\hat{\rho}_k] = \rho_k$.

3. **Classical Shadow:** Collecting $M$ independent snapshots yields

$$\mathcal{S}(\rho_k; M) = \{\hat{\rho}_k^{(1)}, \ldots, \hat{\rho}_k^{(M)}\},$$

the classical shadow of $\rho_k$.

### B.2 PERFORMANCE GUARANTEES AND SAMPLE COMPLEXITY

A key advantage of the classical shadow is its rigorous sample-efficiency guarantee (Huang et al., 2020). Consider predicting expectation values of $L$ observables $\{O_i\}_{i=1}^{L}$ up to additive error $\epsilon$. The number of measurements $M$ required satisfies

$$M \geq \mathcal{O}\left(\frac{\log(L)}{\epsilon^2} \max_i \|O_i\|_{\text{shadow}}^2\right).$$

Here, $\|O\|_{\text{shadow}}$ is the shadow norm associated with the measurement ensemble.

Two key implications are particularly relevant for our setting:

- **System-size independence.** The bound does not scale with the total number of qubits $n$, but only with $\epsilon$, $\log(L)$, and the shadow norms. This makes shadow-based estimation scalable to large systems.

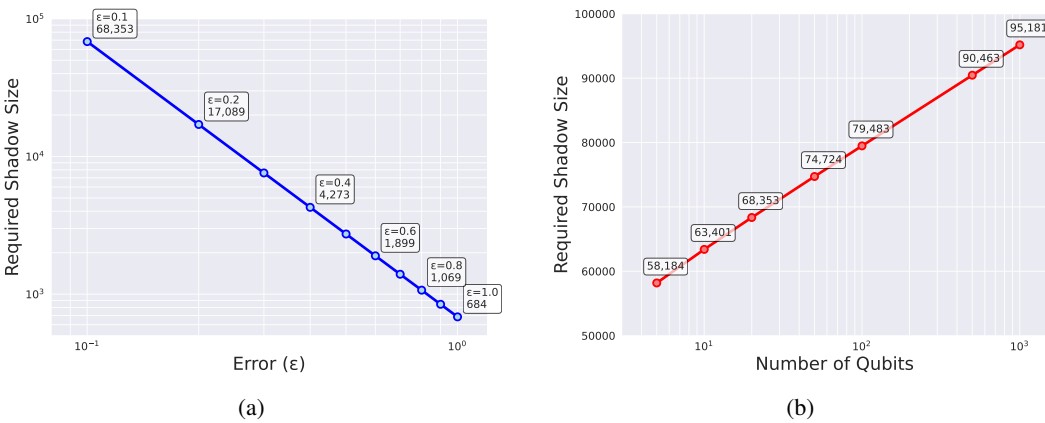

(a)  (b)

Figure 11: Sample complexity of classical shadows. (a) Shadow size required to achieve a target accuracy $\epsilon$ for a 20-qubit system. (b) Scaling of the shadow size with the number of qubits $n$ at a fixed accuracy of $\epsilon = 0.1$.

- **Favorable scaling for local observables.** For Pauli measurements, any $k$-local observable obeys

$$\|O\|_{\text{shadow}}^2 \leq 4^k \|O\|_\infty^2,$$

  ensuring that few-body terms can be estimated with polynomial sample complexity. Since physical Hamiltonians are composed of such local terms, estimating their energies remains efficient.

We evaluated the number of shadow size required by this bound for observables relevant to Krylov diagonalization. First, Figure 11a shows the dependence of the required shadow size on the error tolerance $\epsilon$ for a 20-qubit system. As theoretically predicted, the sample complexity scales as $\mathcal{O}(1/\epsilon^2)$, ranging from $\sim 700$ samples at $\epsilon = 1.0$ to $\sim 6.8 \times 10^4$ at $\epsilon = 0.1$. Second, Figure 11b shows how the sample complexity scales with the number of qubits at a fixed precision of $\epsilon = 0.1$. The required number of shadow size grows logarithmically with the number of qubits. Taken together, these results demonstrate that shadow-based estimation is both theoretically well-founded and practically feasible for large-scale Krylov methods.

### B.3 COMPARISON OF CLASSICAL SHADOW AND GENKSR

Classical Shadow (CS) and GenKSR work under fundamentally different objectives and quantum-resource cost models. CS require new quantum measurements for every Hamiltonian and every Krylov subspace dimension. The CS results in Table 1 assume thousands of shots for each test Hamiltonian and for each Krylov dimension $D = 1, \ldots, 15$. This accumulates substantial quantum cost. In contrast, GenKSR requires quantum measurements only once. GenKSR is trained on shallow-depth data ($D \leq 5$), after which all evaluations—on unseen Hamiltonians and for larger $D$—are performed entirely classically with zero additional QPU shots. The purpose is not to exceed CS under equal-shot budgets, but to eliminate repeated quantum sampling altogether.

The core necessity of the generative model arises from quantum resource savings. GenKSR uses the quantum cost of training data into a one-time expense. In contrast, CS requires repeating quantum circuit executions for every new task. Thus, for workloads involving large Hamiltonian families, repeated diagonalization queries, or exploration across many Krylov dimensions, CS would require significantly more quantum resources, while GenKSR requires none beyond the training stage. Furthermore, GenKSR provides capabilities that CS fundamentally lacks. GenKSR (i) generalizes to unseen Hamiltonians, (ii) extrapolates to larger Krylov dimensions than those used in training, and (iii) reproduces full measurement distributions. CS alone cannot do any of these.

Finally, the performance advantage against real quantum hardware. GenKSR avoids noise accumulation at large Krylov subspace dimensions. While CS can perform well in ideal (noise-free) simulations, it is important to note that large Krylov dimensions require deeper quantum circuits,

which rapidly accumulate noise on real quantum hardware. As a result, CS estimates at large $D$ often suffer from significant hardware-induced errors. In contrast, GenKSR is trained only on shallow-depth circuits (small $D$), where hardware noise is relatively mild, and subsequently extrapolates to larger $D$ entirely classically. This allows GenKSR to bypass the noise accumulation that limits CS at deep circuit depths. This advantage can be seen in our quantum hardware experiment (Figure 5): for example, at Krylov dimension $D = 10$, the Transformer-based GenKSR achieves lower error than direct measurements from the quantum device. This demonstrates that learning from low-noise data and extrapolating classically can outperform noisy large-$D$ quantum circuits in practice.

