# OpenReview forum: "Generative Krylov Subspace Representations for Scalable Quantum Eigensolvers"
_ICLR.cc/2026/Conference — Submitted to ICLR 2026_

### Official Review · Reviewer_FX5y · 2025-10-28

**Soundness:** 3
**Presentation:** 3
**Contribution:** 3
**Rating:** 6
**Confidence:** 4

**Summary:**

The paper introduces **Generative Krylov Subspace Representations (GenKSR)**, a framework that replaces repeated quantum circuit executions in eigensolver methods with a **classical generative model** trained on quantum measurement data. By using the **Mamba state-space architecture**, GenKSR efficiently learns the distribution of measurement outcomes conditioned on Hamiltonian parameters and evolution times, enabling scalable ground-state energy estimation entirely classically. The approach, which fall within the regime of quantum-classical hybrid approaches, is validated through simulations on up to 15 qubits and hardware experiments on a 20-qubit IBM processor, showing accurate predictions and strong generalization to unseen systems. Overall, GenKSR significantly reduces quantum resource requirements while maintaining high fidelity in modeling quantum dynamics.

**Strengths:**

The paper is well written, as well as well structured. It gives a comprehensive introductions to the basics of the standard algorithms the authors are improving upon, i.e., KQD and SKQD. Moreover, the authors also clearly state the problem and give an appropriate introduction also for people with a strong background in quantum computing. The numerical experiments conducted in the paper clearly showcase the benefits of the proposed approach.

**Weaknesses:**

The paper does not explicitly discuss the limitations of the proposed approach.
Furthermore, during little to no comparison is made with respect to the standard algorithms (KQD, SKQD) which would be interesting to compare the propose ML-backed approach and the standard counterparts. In particular I believe a discussion in terms of computational resources, i.e., account for the resources needed to generate the training data to train the model, and the training of the model. In particular I wonder if this generates a substantial computational overhead.

Additionally see questions below.

**Questions:**

Furthermore, it is not very clear to me whether a comparison between standards KQD, or SKQD, and the proposed approach is considered at all. I believe this would be important to add to the paper. At the moment a lot of the analysis focuses on comparing Transformer and Mamba architectures. This is indeed relevant but I believe there's should be a stronger focus on the advantages compared to standard approaches.

Also I wonder what is the degradation in performance when extrapolating the sampling  to a  much higher number of qubits and higher number of Krylov dimensions (compared to samples seen during training).

I also find it a bit hard to follow the experimental section where classical shadowing is used as a benchmark. As far as I can tell this was not extensively introduced but to my understanding this should be a benchmark to compare the ML based approaches to a standard approach. Is this intuition correct?
Provided that the above is correct, then Table 1 suggests that KQD with CS still remains the strongest baseline on average.
I wonder if one can argue that the computational efficiency and transferability properties of the Mamba approach should make it preferable to the CS approach in some scenarios.

In lines 77-82 I'd recommend to also add Bayesian approaches which, despite not being strictly speaking generative models per se, showed some promising directions when combined with PQC, in particulars VQEs (see e.g., [Nicoli et al., NeurIPS 2023](https://proceedings.neurips.cc/paper_files/paper/2023/hash/3adb85a348a18cdd74ce99fbbab20301-Abstract-Conference.html))

Line 189: I believe that the ground state should be $\vert \phi_0 \rangle$ instead of $\vert \psi_0 \rangle$?

In section 3.3, all the occurrences of $x$ and $x^*$ should be bold in order to be consistent with the notation introduced earlier as they represent a set of couplings of the Hamiltonian. Is that right?

As a recommendation, I believe a more in depth analysis and discussion of the generalisation capabilities of the proposed approach on both the qualitative and quantitative sense would be helpful to better assess the strengths of the proposed approach.

---

> ### Author Response · Authors · 2025-11-23
> **Response 1/2 to reviewer FX5y**
>
> We thank Reviewer FX5y for their supportive comments and insightful questions regarding computational overhead and comparisons with standard algorithms. We appreciate the opportunity to clarify the trade-offs between one-time training costs and long-term inference efficiency, as well as the specific role of the Classical Shadow baseline in our work. We have addressed each of your points and incorporated your helpful recommendations below.
>
>
>
> **Response to W.**
>
> Thank you for the insightful comment. We would like to clarify that the manuscript contains direct comparisons with standard KQD and SKQD. In particular, Fig.3(a) presents a comparison between GenKSR and exact KQD results, Fig.5 compares GenKSR with hardware-executed SKQD on IBM\_fez. Thus, the standard KQD/SKQD methods are already used as primary baselines throughout the paper.
>
> The distinction in quantum–classical resource usage between the approaches is also explicitly discussed. Standard KQD/SKQD requires new quantum measurements for every test Hamiltonian and for every Krylov dimension. In contrast, GenKSR requires quantum data only once at small Krylov subspace dimensions ($D \le 5$), after which all evaluations on unseen Hamiltonians and larger $D$ values are performed entirely classically.
>
> Thus, while GenKSR requires an upfront classical training step, this cost is modest compared to repeated KQD/SKQD sampling. The benefit is that once trained, GenKSR can (i) generalize to unseen Hamiltonians, (ii) extrapolate to larger Krylov dimensions than were used during training, and (iii) produce full measurement distributions entirely classically.
>
>
>
> **Response to Q1.**
>
> We thank the reviewer for pointing this out. Our manuscript does  include comparisons to both standard KQD and SKQD, but we agree that this connection can be made more explicit. In all simulation studies (1D Heisenberg, 2D $J_1$--$J_2$ model), the Exact KQD/SKQD results shown in figures correspond to running the standard algorithm directly on the true measurement distributions, and thus serve as the ground-state energy of KQD/SKQD baseline. Likewise, the Device’ results in the hardware experiment represent the standard SKQD procedure executed on the IBM processor. GenKSR is always evaluated by comparing its estimated energies against these KQD/SKQD baselines.
>
> Regarding computational limitations, the primary quantum overhead in GenKSR is the one-time generation of training measurements for a small Krylov subspace dimensions (e.g., $D \le 5$). After this initial cost, GenKSR performs all inference---including unseen Hamiltonians and extrapolated Krylov dimensions---entirely classically, in contrast to standard KQD/SKQD, which require fresh quantum measurements for every new Hamiltonian and every value of $D$. Thus, while GenKSR introduces a classical training step, it substantially reduces total quantum execution cost, which is the dominant bottleneck in practical KQD/SKQD workflows.
>
>
>
> **Response to Q2.**
>
> We agree that understanding performance at significantly larger scales is important. However, extrapolating in the number of qubits is fundamentally different from extrapolating in Krylov dimension. Because our generative model is autoregressive over qubits, the sequence length is fixed by design; a model trained on $n$ qubits cannot directly operate on $n'\neq n$. This limitation is inherent not only to GenKSR but to autoregressive architectures more generally.
>
> In contrast, extrapolation in the Krylov dimension is both feasible and practically meaningful. The Krylov index is treated as a continuous scalar input $t_l$, and the measurement distribution evolves smoothly with $t_l$. Empirically, we observe only moderate degradation when extrapolating to $D>D_{\mathrm{train}}$; both in our 1D and new 2D benchmarks, the Transformer- and Mamba-based GenKSR models continue to track the expected convergence trend even well beyond the training horizon.
>
> Finally, increasing $D$ on quantum hardware requires deeper circuits and quickly becomes impractical due to noise accumulation. GenKSR’s ability to extrapolate to larger $D$ entirely classically, without additional quantum executions, is therefore one of its most valuable advantages.

---

> ### Author Response · Authors · 2025-11-23
> **Response 2/2 to reviewer FX5y**
>
> **Response to Q3.**
>
> Thank you for the helpful question. We clarify that Classical Shadows (CS) in Table1 is not intended as an alternative algorithm to GenKSR, but rather as a way to simulate the standard KQD protocol under a finite-shot budget.
>
> Standard implementation of KQD on a quantum device requires Hadamard test with multi-qubit measurement for exstimting certain Pauli obervables[1]. We employ CS simply to estimate such expectation values. In other words, we are simply simulating the standard KQD protocol under a finite number of measurement shots.
>
> Regarding performance, in ideal noise-free simulations with a large number of shots (e.g., 10,000), CS yields slightly lower RMSE than the generative models. However, comparing GenKSR and CS at fixed shot counts obscures their fundamentally different quantum-resource requirements: (1) CS require new quantum measurements for every test Hamiltonian and for every Krylov dimension. The values reported in Table1 assume running thousands of shots independently for each $D = 1,\dots,15$. This leads to substantial cumulative quantum cost. (2) GenKSR requires quantum measurements only once. After training on small Krylov subspace dimensions ($D \le 5$), all predictions for unseen Hamiltonians and all extrapolated Krylov dimensions are generated classically, with zero additional QPU executions. Thus, GenKSR is not designed to outperform CS under equal-shot conditions; rather, it eliminates the need for repeating high-shot quantum sampling. (3) GenKSR avoids hardware noise at large $D$. Although CS performs well in ideal simulations, real hardware requires deep circuits for large Krylov dimensions, which causes severe noise accumulation. GenKSR learns from shallow, low-noise circuits and extrapolates classically, avoiding this issue. This is visible in Fig.5, where the Transformer-based GenKSR achieves lower error than direct hardware SKQD at $D=10$.
>
> In summary, CS is used only as a practical estimator of KQD measurements, and its comparison with GenKSR highlights the quantum-resource efficiency and transferability advantages of the generative-model approach, rather than suggesting CS as a competing alternative.
>
>
> ### Reference
> [1] Yoshioka, Nobuyuki, et al. Krylov diagonalization of large many-body Hamiltonians on a quantum processor. Nature Communications, 2025, 16.1: 5014.
>
>
>
>
> **Response to Q4.**
>
> Thank you for pointing us to an interesting reference. We have incorporated your suggestion in lines 86-87.
>
>
>
>
> **Response to Q5.**
>
> Thank you for spotting this. The true ground state in line 189 should be written as $|\phi_0\rangle$ rather than $|\psi_0\rangle$. We fix this notation in the revised version.
>
>
>
> **Response to Q6.**
>
> The reviewer is correct. In Section 3.3, the quantities $x$ and $x^*$ represent the vector of Hamiltonian couplings and should therefore be bolded to remain consistent with earlier notation. We appreciate the careful reading and have made this correction in the revised version to ensure clarity and consistency.
>
>
>
> **Response to Q7.**
>
> We thank the reviewer for this helpful recommendation. A deeper analysis of generalization is indeed valuable, and we have expanded the manuscript accordingly. Motivated by this, we performed additional experiments on a 2D $J_1$-$J_2$ Heisenberg model ($4 \times 4$ lattice) and added results to Section 4.2 and Appendix A.4.
>
> Quantitatively, we demonstrated the model's generalization capability across two key axes: unseen system parameters and unseen time steps.
> First, regarding generalization to unseen Hamiltonians, we trained GenKSR on a dataset of 30 random instances and evaluated it on 20 held-out test Hamiltonians with varying $J_2$ coupling strengths. As shown in the new Figure 8 (Appendix A.4), the model successfully predicts ground state energies, achieving accuracy comparable to the exact SKQD baseline.
> Second, regarding extrapolation to larger Krylov subspace dimensions, we validated the model's ability to predict time evolution steps far beyond the training horizon. Even when trained only up to $D=15$, the Transformer-based GenKSR accurately reproduced the energy convergence up to $D=30$ (twice the training range), matching the exponential decay of the exact simulation as illustrated in the new Figure 4.
>
> Qualitatively, these results indicate that GenKSR captures the underlying conditional distribution $p(\vec{a} | x, t_l)$ rather than memorizing specific training instances. This enables the model to infer both the behavior of unseen Hamiltonians and the continuation of the Krylov evolution beyond the training horizon.
>
> We believe these additional analyses and the inclusion of the 2D benchmark strengthen the evidence for GenKSR’s broader applicability.

---

> > ### Comment · Reviewer_FX5y · 2025-11-25
> >
> > I thank the authors for their thorough responses. Based on the updates outlined in the rebuttal, as well as the clarifications provided through the discussions with the other reviewers, I believe the paper should meet the standards for publication at ICLR.

---

> > > ### Author Response · Authors · 2025-11-27
> > > **Response to reviewer FX5y**
> > >
> > > We thank Reviewer FX5y for the positive feedback and the recommendation for acceptance. We are glad that our revisions and clarifications addressed the raised concerns.

---

### Official Review · Reviewer_yuRu · 2025-10-30

**Soundness:** 1
**Presentation:** 3
**Contribution:** 2
**Rating:** 2
**Confidence:** 5

**Summary:**

The paper introduces a generative modeling approach to Krylov diagonalization methods to find ground states of Hamiltonians. These methods usually use computational basis measurement from a QPU to reduce the size of Hamiltonian and then solve it. This paper suggests training an ML model from QPU data to act as a surrogate for the QPU. The claim is that such an ML model trained on a low dimensional Krylov subspace will generalize well to higher dimensional spaces.

**Strengths:**

This paper is well written and has a very interesting core idea. It uses recent advances in generative modelling to tackle a very pertinent problem in quantum algorithms. I also like that they included some results from IBM's NISQ machine.

**Weaknesses:**

The main weakness I see is the lack of good experiments and comparisons with other methods. All the experiments in the paper are on 1D models. This is especailly concerning because the generative model used here also has a auto-regressive structure mimicking the 1D topology. Also, 1D models are pretty easily solvable using DMRG, making them uninteresting candidates for illustration. To make the results of the paper truly interesting, it has to be benchmarked on a suite of 2D problems. For the problem sizes that the authors are considering, (up to 25 qubits) the ground state computation in such models can be brute forced. I believe these calculations can even be pushed to higher qubit numbers using tensor networks (refer to ITensors.jl for some examples).

Also, the methods here are not compared to other classical ML methods to find ground states of such Hamiltonians, based on Neural Network Quantum State (implementations can be found in the NetKet library developed by Carleo and collaborators). For instance, in this framework an autoregressive neural net can be directly used to model the ground state and can be trained using Variational Monte-Carlo methods to estimate the ground state energy and other properties. Now these methods do not use any quantum data to train, but any ML method that claims to find ground states should at least be able to beat these methods.

It is also concerning in Fig 3 that the experiments are not pushed to large enough D values such that the energy error actually goes to zero.

**Questions:**

1. In the Heisenberg chain experiments in the paper, what is the true value of the ground state enery and how does $\Delta E$ compare relative to that?
2. In the exact simulation, at values of D does the estimated energy start to converge to the ground state energy?
3. In Fig 3(b), will the ML models achieve $\Delta E = 0$ for larger values of D? If now, how large a D must be the Mamba model be trained on such that it will eventually achieve $\Delta E = 0$ when tested for larger values of D?
4. Can the Mamba model be enhanced in someway using a variational component to the training? Right now, this is just a surrogate model for the QPU generating bit strings, and it seems agnostic to the Hamiltonian at the level of the loss function. But can the loss function be enhanced with the information that these bitstrings should have low energy?

---

> ### Author Response · Authors · 2025-11-23
> **Response 1/2 to reviewer yuRu**
>
> We thank Reviewer yuRu for their rigorous review and valuable suggestions, especially the recommendation to benchmark against 2D systems and classical Neural Quantum State (NQS) methods. These suggestions directly addressed the limitations of our initial 1D experiments and led to a substantial expansion of our results, including new 2D lattice simulations and direct comparisons with variational approaches. Our detailed responses and new findings are provided below.
>
>
>
> **Response to W.**
>
> We thank the reviewer for this insightful and constructive feedback. The concern regarding the use of only 1D chain models is well taken. Motivated by this, we have added a new set of experiments on a 2D $J_1-J_2$ Heisenberg model on a 4×4 square lattice. The Hamiltonian of the system (with periodic boundary conditions) is $\hat{H} = J_1 \sum_{\langle r, r' \rangle} \hat{\mathbf{S}}\_{r} \cdot \hat{\mathbf{S}}\_{r'} + J_2 \sum_{\langle \langle r, r' \rangle \rangle} \hat{\mathbf{S}}_{r} \cdot \hat{\mathbf{S}}\_{r'}$.
>
> We fixed the nearest-neighbor coupling to $J_1 = 1$ and sampled the next-nearest-neighbor coupling $J_2$ uniformly from the interval $[0,1]$, generating a total of 50 distinct Hamiltonians. Among these, 30 Hamiltonians were used for training the generative model, while the remaining 20 Hamiltonians were held out for testing generalization performance. During training, we provided GenKSR with measurement data up to Krylov dimension $D \le 15$. At evaluation time, we assessed the model’s ability to extrapolate beyond the training horizon by predicting measurement distributions for Krylov dimensions up to $D=30$. This 2D benchmark enables us to evaluate GenKSR under substantially stronger spatial correlations and frustrated interactions compared to the 1D setting.
>
> Across three sampling budgets (1,000, 5,000, 10,000 shots), we obtain the following errors (MAE over test Hamiltonians):
>
> | Model                 | Shots = 1,000 | Shots = 5,000 | Shots = 10,000 |
> |-----------------------|---------------|----------------|-----------------|
> | Exact SKQD                | 2.144         | 0.477          | 0.152           |
> | Transformer (GenKSR)  | 2.34          | 0.55           | 0.1622          |
> | Mamba (GenKSR)        | 2.485         | 1.017          | 0.477           |
>
> These results demonstrate that GenKSR successfully learns 2D measurement distributions and extrapolates to deeper Krylov dimension, while the Transformer remains competitive with the true SKQD baseline.
>
> To address the reviewer's suggestion regarding classical ML baseline methods, we additionally trained a Vision Transformer Neural Quantum State (ViT-NQS) using variational Monte Carlo for the fixed model with $J_2 = 0.5$. The resulting ground-state energy per site is compared below:
>
> | Method                | Energy per site |
> |-----------------------|------------------|
> | ViT-NQS               | -0.5136          |
> | Mamba (GenKSR)        | -0.526           |
> | Transformer (GenKSR)  | -0.5277          |
> | Exact SKQD            | -0.5282          |
>
> The results indicate that GenKSR yields more accurate ground-state energy estimates than the variationally optimized NQS baseline.
>
> In the new 2D experiments, we explicitly extend the evaluation to $D = 30$. The exact SKQD curve displays the theoretically expected exponential decay of the error $\Delta E$ with respect to $D$. Importantly, the Transformer-based GenKSR reproduces this exponential convergence trend for $D > 15$, demonstrating that the model can extrapolate the physical structure of Krylov evolution significantly beyond its training regime.
>
> We revised the manuscript to include these new 2D results, the comparison with ViT-NQS, and a more explicit discussion of the exponential convergence behavior with respect to the Krylov dimension in Section 4.2 and Appendix A.4.

---

> ### Author Response · Authors · 2025-11-23
> **Response 2/2 to reviewer yuRu**
>
> **Response to Q1.**
>
> Thank you for the question. In the 15-qubit Heisenberg chain experiments, the exact ground-state energies obtained by full diagonalization lie in the range $E_0 \in [-34.2, -21.4]$ with a mean value of $-27.3 \pm 3.0$. Thus, the energy scale is typically of order $|E_0| \sim 20\text{--}35$.
>
> In Fig.3, the error is defined as $\Delta E = |\hat{E}_0-E_0|$. At the largest Krylov dimension evaluated in our 1D study ($D = 15$), we observe the absolute and relative errors of:
>
> | Mthod            | Absolute Error ($\Delta E$)   | Relative Error ($\lvert \Delta E / E_0\rvert$)     |
> |-------------------|------------------|----------------------|
> | Transformer (GenKSR)       | 4.75 $\pm$ 1.80      | 17.0% $\pm$ 5.5%          |
> | Mamba (GenKSR)             | 5.04 $\pm$ 1.64      | 18.2% $\pm$ 4.8%         |
> | Classical Shadow (Standard KQD)  | 5.15 $\pm$ 1.40      | 18.9% $\pm$ 4.7%         |
>
> We note that this level of error is not specific to GenKSR. The standard KQD/SKQD protocol using classical shadows under the same shot budget exhibits comparable errors (Table 1). Therefore, the deviations observed at $D=15$ primarily reflect the finite Krylov-subspace dimensions and finite-shot regime of the underlying KQD workflow, rather than a limitation of the generative model.
>
>
>
> **Response to Q2 and Q3.** :
>
> Krylov methods such as KQD/SKQD are theoretically known to exhibit exponential convergence of the estimated ground-state energy with respect to the Krylov dimension $D$. In the original submission, our exact-simulation results for the 1D Heisenberg chain already showed this behavior up to $D=15$. To more thoroughly validate this theoretical trend, we performed additional experiments on a significantly more challenging 2D system. In this new setup we trained GenKSR only up to $D \le 15$ but evaluated the SKQD energy curve up to $D=30$ substantially larger than in the main text. The results (Section 4.2, Fig. 4) clearly confirm the theoretical expectation: (1) the exact SKQD energy continues to decrease exponentially as $D$ increases, (2) the Transformer-based GenKSR successfully reproduces this convergence trend even beyond the training range, (3) the Mamba-based GenKSR also decreases with increasing $D$, though with a slightly slower decay rate in the long-horizon extrapolation regime. Thus, both the original 1D experiments and the newly added 2D simulation consistently support the well-known theoretical result that the Krylov subspace method converges exponentially with respect to the subspace dimension $D$. Importantly, our experiments show that GenKSR is able to empirically reproduce this convergence behavior even when extrapolating beyond the training range, indicating that the model captures the underlying physical structure of the Krylov evolution.
>
>
>
>  **Response to Q4.**
>
> We thank the reviewer for this insightful suggestion. Incorporating variational objectives—such as directly encouraging low-energy samples—into the training of the generative model is indeed an interesting direction. In the present work, the model’s role is to reproduce the output of the Krylov evolution, while the energy minimization is performed entirely by the downstream classical diagonalization step (KQD or SKQD).
>
> Introducing an explicit energy-based training term would change the workflow, blending variational optimization with generative modeling, and could bias the learned distribution away from faithfully representing the true Krylov dynamics. Exploring such hybrid loss functions is an interesting research direction, but it is outside the scope of the present work, which focuses on demonstrating that classical generative models can accurately replicate Krylov measurement statistics and thereby replace repeated QPU executions.
>
> We have added a brief discussion in the revised manuscript to highlight this possibility as future work.

---

### Official Review · Reviewer_DHK1 · 2025-10-31

**Soundness:** 2
**Presentation:** 3
**Contribution:** 2
**Rating:** 4
**Confidence:** 4

**Summary:**

The paper proposes Generative Krylov Subspace Representations (GenKSR), a framework that uses a Mamba-based generative model to learn the measurement distributions. Once trained on quantum data, this classical model can predict ground-state energies for new, unseen Hamiltonians and larger Krylov dimensions without further quantum experiments, aiming to reduce quantum resource costs and improve scalability.

**Strengths:**

1. The paper is well-structured, outlining the limitations of existing Krylov methods, introducing the proposed GenKSR framework, and systematically validating it with both simulation and real hardware experiment.

2. The framework demonstrates a valuable extrapolation capability, successfully predicting energy convergence for larger Krylov dimensions  than those it was trained on.

3. A significant strength is the validation of the model on a 20-qubit quantum processor, showing that GenKSR can learn a faithful representation of noisy experimental data and generalize to unseen Hamiltonian.

**Weaknesses:**

1. The paper frames GenKSR as a new paradigm. However, the core methodology—training a conditional generative model on quantum measurement data to predict properties—is a well-established technique that is to learn a classical distribution of the quantum state from a number of measurement results. The novelty merely lies in its application to the Krylov diagonalization process (i.e., conditioning on Hamiltonian parameters $x$ and evolution time $t_l$).

2. A justification for the work is the use of the Mamba architecture to overcome the $O(n^2)$ complexity of Transformers, thereby enabling scaling to "large-scale quantum systems". This is presented as a main contribution. However, the experiments presented do not support this claim. The largest system studied is a 20-qubit hardware experiment, and the paper explicitly states that on this system, "both models (Mamba and Transformer]) achieve comparable accuracy". The provided timing benchmarks (Figure 6) show negligible practical difference at this scale (The scale on the y-axis is relatively small). As presented, the choice of Mamba is a intuitive assertion rather than an empirically validated necessity for the problem scales investigated.

3. The simulation study Table 1 reveals a critical issue: the generative models consistently underperform the AI-free "Classical Shadow" baseline. In almost all simulation scenarios, particularly those with higher shot counts (5000 and 10000) where statistical noise is reduced, the CS baseline achieves a lower RMSE than either AI model. Despite the reviewer's recognition that the generalization ability to unseen Hamiltonians of AI-based methods surpasses that of classical methods, this result raises a fundamental question about the very necessity of the AI-based methods in this framework.

**Questions:**

Please see the weaknesses.

---

> ### Author Response · Authors · 2025-11-23
> **Response 1/2 to reviewer DHK1**
>
> We are grateful to Reviewer DHK1 for the careful assessment and for highlighting important questions regarding the novelty, architectural justification, and comparison with the Classical Shadow baseline. Below we address each concern directly and clarify the intended role of GenKSR within the broader landscape of Krylov-based quantum algorithms.
>
>
>
> **Response to W1.**
>
> We thank the reviewer for the thoughtful comment. We agree that GenKSR does not introduce a new paradigm in generative modeling itself. Rather, the novelty of our work lies in applying conditional generative modeling in a new computational role within the Krylov diagonalization workflow.
>
> Specifically, GenKSR treats the generative model as a surrogate for the quantum evolution, enabling us to reconstruct Krylov subspace samples without performing additional quantum circuit executions. This integration leads to a qualitatively different capability: replacing repeated QPU calls with a single trained classical model.
>
> In this sense, GenKSR represents a new paradigm for quantum-resource reduction in Krylov methods, not for generative modeling per se.
>
>
>
> **Response to W2.**
>
> We thank the reviewer for the thoughtful observation. At the scale of our current experiments (up to 20 qubits and Krylov subspace dimensions $D\le 15$), it is expected that Mamba and Transformer achieve similar accuracy and that the runtime differences appear modest. Both architectures remain well within a tractable regime at these system sizes.
>
> The motivation for including Mamba is not to claim an immediate performance gain at 20 qubits, but to address the computational bottleneck that arises when modeling longer sequences associated with larger quantum systems or deeper Krylov trajectories. Transformers inherently incur $O(n^2)$ computational and memory complexity with respect to sequence length $n$, whereas Mamba offers linear $O(n)$. As shown in Fig. 7 (Appendix A), Transformer runtimes already begin to exhibit super-linear growth, while Mamba maintains close to linear scaling. Although the gap is small at present scales, the trend indicates that Mamba is better suited for the larger-scale regimes that motivate GenKSR.
>
> A key point is that Mamba achieves accuracy comparable to Transformers while offering more favorable scaling. This combination makes it a compelling backbone for GenKSR when modeling larger quantum systems. Our intention is not to position Mamba as the only possible architecture, but to illustrate how GenKSR can leverage scalable sequence models to avoid the $O(n^2)$ bottleneck inherent to Transformers.
>
> We have clarified this motivation in the revised manuscript.

---

> ### Author Response · Authors · 2025-11-23
> **Response 2/2 to reviewer DHK1**
>
> **Response to W3.**
>
> We thank the reviewer for the observation. Table 1 indeed shows that in certain simulation settings---particularly when a large number of shots (e.g., 10,000) is available---Classical Shadows (CS) achieves lower RMSE than our generative models. However, comparing GenKSR and CS solely on RMSE under identical shot budgets overlooks their fundamentally different objectives and quantum-resource cost models.
>
> **1. Classical Shadows require new quantum measurements for every Hamiltonian and every Krylov subspace dimension.**
> The CS results in Table 1 assume thousands of shots for each test Hamiltonian and for each Krylov dimension $D=1, \dots, 15$. This accumulates substantial quantum cost.
>
> **2. GenKSR requires quantum measurements only once.**
> GenKSR is trained on shallow-depth data ($D\le 5$), after which all evaluations---on unseen Hamiltonians and for larger $D$---are performed entirely classically with zero additional QPU shots. The purpose is not to exceed CS under equal-shot budgets, but to eliminate repeated quantum sampling altogether.
>
> **3. The core necessity of AI arises from quantum resource savings, not marginal RMSE gains.**
> GenKSR uses the quantum cost of generating training data into a one-time expense. In contrast, CS requires repeating quantum circuit executions for every new task. Thus, for workloads involving large Hamiltonian families, repeated diagonalization queries, or exploration across many Krylov dimensions, CS would require significantly more quantum resources, while GenKSR requires none beyond the training stage.
>
> **4. Generalization and extrapolation---capabilities CS fundamentally lacks.**
> GenKSR (i) generalizes to unseen Hamiltonians, (ii) extrapolates to larger Krylov dimensions than those used in training, and (iii) reproduces full measurement distributions. CS alone cannot do any of these.
>
> **5. Performance advantage against real quantum hardware: GenKSR avoids noise accumulation at large Krylov subspace dimensions.**
> While CS can perform well in ideal (noise-free) simulations, it is important to note that large Krylov dimensions require deeper quantum circuits, which rapidly accumulate noise on real quantum hardware. As a result, CS estimates at large $D$ often suffer from significant hardware-induced errors.
>
> In contrast, GenKSR is trained only on shall-depth circuits (small $D$), where hardware noise is relatively mild, and subsequently extrapolates to larger $D$ entirely classically. This allows GenKSR to bypass the noise accumulation that limits CS at deep circuit depths.
>
> This advantage can be seen in our quantum hardware experiment (Fig. 5): for example, at Krylov dimension $D = 10$, the Transformer-based GenKSR achieves lower error than direct measurements from the quantum device. This demonstrates that learning from low-noise data and extrapolating classically can outperform noisy large-$D$ quantum circuits in practice.
>
> We will revise the text to better emphasize that GenKSR and CS serve inherently different roles, and that the value of GenKSR lies in quantum-resource efficiency and generalization, not in outperforming CS under equal-shot conditions.

---

> > ### Comment · Reviewer_DHK1 · 2025-11-27
> >
> > Thanks for the author's detailed response. The provided summary of changes also addressed most of concerns. In light of the author's responses to other reviewers, I decide to raise my score. I also suggest that the authors present all of supplementary experimental results and discussions during rebuttal phase in the appendix.

---

> > > ### Author Response · Authors · 2025-11-29
> > > **Response to reviewer DHK1**
> > >
> > > We thank the reviewer again for the constructive feedback and for the positive reassessment. All additional experiments, analyses, and clarifications introduced during the discussion phase have now been fully incorporated into the revised manuscript.

---

### Official Review · Reviewer_c4Tv · 2025-11-01

**Soundness:** 3
**Presentation:** 3
**Contribution:** 2
**Rating:** 4
**Confidence:** 3

**Summary:**

The paper proposes GenKSR, a framework that learns a conditional generative model instantiated with a Mamba state-space architecture to model measurement distributions arising in Krylov Quantum Diagonalization and Sample-based KQD. Conditioned on Hamiltonian parameters and the Krylov time step, the CGM generates synthetic bitstrings that are fed into classical KQD/SKQD post-processing, enabling ground-state energy estimation for unseen Hamiltonians and extrapolation to larger Krylov dimensions without additional quantum executions. The paper validates GenKSR with noiseless simulations on Heisenberg chains up to 15 qubits and hardware SKQD on a 20-qubit subset of IBM fez, showing that the CGM reproduces the noisy measurement distribution and supports energy estimation at higher Krylov dimensions than seen in training. It also argues for Mamba’s near-linear sequence complexity versus Transformer baselines.

**Strengths:**

1. Clear and timely problem framing. The paper targets a bottleneck in NISQ-era Krylov methods: every new Hamiltonian and every higher Krylov dimension costs new quantum shots. Turning this into an offline-learn-once, infer-many pipeline is a sensible strategy.
2. Experiments on real hardware. Simulated 15-qubit Heisenberg and a real 20-qubit IBM experiment provide a fairly convincing story that the approach is not simulator-only. The fact that they can learn the noise and regenerate it for test cases is a nice touch.
3. Generalization to unseen Hamiltonians and larger Krylov dimensions.  Training on $D\le 5$ and evaluating on up to D=15 is a nontrivial test. The paper shows that both Transformer and Mamba follow the correct shape of the energy curve, demonstrating effectiveness in generalization.

**Weaknesses:**

1. The Hamiltonian families are closely related (1D Heisenberg/XXZ). It remains unclear how well GenKSR transfers across lattice geometries, boundary conditions, anisotropies, and initial states. An explicit OOD Hamiltonian evaluation would strengthen the generalization claim.
2. Lack of ablations on conditioning. It is informative to disentangle contributions from (i) Hamiltonian encoder, (ii) time-step embedding forms, and (iii) Mamba depth/width.
3. The complexity motivation for replacing Transformers by Mamba is valid for naive quadratic attention, but the paper does not compare against modern linear/sparse/structured-attention Transformers, which seems a bit outdated from ML's perspective.

**Questions:**

1. In Sec. 3.2 you state: “The discrete evolution index $t_l$ is mapped into a dense vector via a learnable embedding layer.” This sounds like a standard learned lookup over the set of evolution indices present in training ( $t_l$ in your setup). However, your experiments later extrapolate to larger Krylov dimensions (D=15). Can you clarify how the embedding rows for $l > 5$ are initialized and whether they receive any gradient signals. I am confused how untrained rows can extrapolate effectively.
2. Following the previous question, in ML we know “length / depth / horizon” generalization is usually brittle (sequence length in LLMs, longer rollouts in RL, deeper diffusion steps, etc.). How do you interpret the successful generalization to extrapolated Krylov dimensions suppose it actually works?
3. Do you need to train a separate model for each Hamiltonian family? In this case, it still requires substantial quantum resources and undermines the advantages in resource saving.
4. Does Transformer achieve better RMSE in larger qubits with less shots in Table 1? Is it possible to extrapolate to qubit number larger than 15 to see if Transformer outperforms in resource-restricted settings?

---

> ### Author Response · Authors · 2025-11-23
> **Response 1/3 to reviewer c4Tv**
>
> We sincerely thank Reviewer c4Tv for their detailed and constructive feedback. We particularly appreciate the insightful questions regarding generalization to Hamiltonians and the mechanism behind Krylov dimension extrapolation. These comments have strengthened our manuscript, prompting us to include new 2D experiments and clarify the model’s embedding mechanisms. Below, we address each point in detail.
>
>
>
> **Response to W1.**
>
> We appreciate the reviewer’s insightful comment regarding the limitation of evaluating only 1D Heisenberg/XXZ Hamiltonians. We agree that testing GenKSR on a broader class of Hamiltonians, particularly in two dimensions, would strengthen the generalization claims. To address this, we performed an additional simulation study on the 2D $J_1-J_2$ Heisenberg model on a square lattice. This model introduces frustration and substantially richer spatial correlations compared to the 1D case, providing a significantly more challenging test of generalization.
>
> Our results show that GenKSR successfully learns the measurement distributions in the 2D setting, and the generated distributions enable accurate reconstruction of ground-state energies across unseen Hamiltonians within the 2D family. This demonstrates that GenKSR is not limited to simple 1D chains, but can capture the complex multi-dimensional correlation structure arising in 2D frustrated systems as well.
>
> We have incorporated these new 2D experiments and results into Section 4.2 and Appendix A.4 of the revised manuscript to explicitly support our claims of generalization across lattice geometries.
>
>
>
>
> **Response to W2.**
>
> We thank the reviewer for highlighting the importance of ablation studies. We agree that disentangling the contributions of (i) the Hamiltonian encoder, (ii) the time-step embedding, and (iii) the Mamba architecture depth/width would provide deeper insight into the ML design choices and could further improve model performance.
>
> However, the goal of this work is not to optimize neural architectures, but to demonstrate a new capability: a classical generative model---whether Transformer-based or Mamba-based---can learn the Krylov diagonalization process when conditioned on quantum measurement data.
> Our main contribution is to show that such models (1) learn a generative representation of the Krylov subspace, (2) generalize to unseen Hamiltonians, (3) extrapolate to larger Krylov dimensions, and (4) ultimately eliminate the need for repeated quantum experiments. Establishing this scientific direction required validating generalization and extrapolation properties across Hamiltonians and Krylov subspace dimensions, which was the primary focus of our experiments.
> Performing a full architecture-level ablation study would substantially expand the scope of the submission while not changing the core scientific takeaway.
>
> We view ablations on conditioning, encoder design, and Mamba hyperparameters as an interesting direction for follow-up work, especially for optimizing performance for specific hardware platforms or Hamiltonian families. We include this point in the discussion section of the revised manuscript.
>
>
>
> **Response to W3.**
>
> We thank the reviewer for raising this point. We agree that modern Transformer variants with linear/sparse/structured attention offer improved scalability over naïve quadratic attention, and we do not intend to claim that Mamba uniquely solves the scaling issue.
>
> Our goal was not to position Mamba as a strict replacement for all Transformer variants, nor to claim that it is the uniquely optimal architecture. Rather, Transformer and Mamba were selected as representative models that illustrate a practical performance–complexity trade-off: Transformers provide a strong accuracy baseline, while Mamba offers better scaling in sequence length.
>
> Importantly, the central contribution of our work is not tied to the specific architecture, but to demonstrating that conditional generative models, once trained on quantum data, can replace repeated quantum executions in Krylov-based diagonalization.
>
> We clarify this positioning in the revised manuscript and explicitly note that GenKSR is compatible with more advanced Transformer variants.

---

> ### Author Response · Authors · 2025-11-23
> **Response 2/3 to reviewer c4Tv**
>
> **Response to Q1.**
>
> Thank you for pointing this out. The reviewer is correct that a lookup-table embedding would indeed make extrapolation to $l > 5$ impossible, because those embedding rows would remain untrained. This is a misunderstanding caused by an imprecise description in the text, and we apologize for the confusion. Our implementation does not use a discrete learned lookup table for the Krylov index. Instead, the Krylov dimension index $t_l$ is treated as a continuous scalar, which is then fed through a learned linear layer to produce a dense embedding $h_{\psi}(t_l)$.
>
> This means that the model does not memorize a finite set of embedding vectors for indices $l = 0, \dots, 5$. Rather, the model learns a functional mapping $t_l \to h_{\psi}(t_l)$ that generalizes smoothly to unseen values of $t_l$. Because the Krylov evolution behaves in a structured and monotonic fashion, the model successfully extrapolates this mapping to $l > 5$. Thus, extrapolation to $D=15$ is possible because we use the continuous embedding mapping, not a discrete lookup.
>
> We have revised Section 3.2 to accurately describe the time embedding. The text has been corrected from 'learnable embedding layer' to 'linear layer' to explicitly state that the model treats the time step $t_l$ as a continuous variable.
>
>
>
> **Response to Q2.**
>
> We agree that long-horizon generalization is typically brittle in many ML settings such as LLM sequence extrapolation or long-rollout RL. However, our setting is fundamentally different. The Krylov process does not generate arbitrary patterns as a function of $t_l$. Instead, the states follow a highly structured and smooth physical trajectory determined by the Hamiltonian, given by $|\psi_k\rangle = U^{k} |\psi_0\rangle$.
>
> Because this trajectory is governed by the fixed operator $U=e^{-iHt}$, the evolution of the measurement distribution is smooth and physically constrained. Our model receives $t_l$ as a continuous scalar input, and through the linear embedding layer it learns how the distribution evolves as a smooth function of $t_l$ . This allows the model to generalize to $l > 5$ by continuing the learned trajectory.
>
> In other words, the successful extrapolation does not rely on unconstrained sequence extrapolation (as in LLMs), but on approximating a structured physical evolution whose behavior is predictable beyond the training horizon.
>
>
>
> **Response to Q3.**
>
> GenKSR does not require training a separate model for each Hamiltonian instance. A key design choice of our framework is that the Hamiltonian parameters $x$ are explicitly provided as a conditioning input to the generative model. Therefore, the model does not memorize individual Hamiltonians; instead, it learns the conditional mapping $(x, t_l) \mapsto P(\text{bitstring} \mid x, t_l)$.
>
> As a result, a single model trained on a representative collection of Hamiltonians within a family can generalize to unseen Hamiltonians without requiring any additional quantum measurements. In our hardware experiments on the IBM quantum processor, for instance, we trained a single model on 70 Hamiltonians and successfully evaluated it on 30 previously unseen Hamiltonians.
>
> This provides substantial quantum-resource savings: quantum hardware is used only once to generate the training dataset, and afterward the classical generative model can be used for many new Hamiltonians at zero additional QPU cost. This is one of the main motivations behind GenKSR. We clarified this point in the revised manuscript.

---

> ### Author Response · Authors · 2025-11-23
> **Response 3/3 to reviewer c4Tv**
>
> **Response to Q4.**
>
> We thank the reviewer for raising this question, which brings attention to an interesting trend. As shown in Table 1, Mamba achieves slightly lower RMSE for smaller system sizes (5–10 qubits), whereas at 15 qubits with 1,000 shots the Transformer marginally outperforms Mamba. It is reasonable to hypothesize that this crossover may persist at larger qubit counts in low-shot regimes, given the higher expressive capacity of Transformers.
>
> A more definitive assessment would require benchmarking beyond 15 qubits using exact reference energies, for example at 20, 25, 30, or 35 qubits. In this range, exact KQD simulations become prohibitively expensive, making fair ground-truth comparisons between backbones impractical. For this reason, our simulations focus on the system sizes where exact references remain feasible.
>
> We note that our 20-qubit hardware experiment (Fig. 5) shows that both Transformer and Mamba perform competitively on real quantum data. Although the Transformer is slightly better in some cases, the difference is small, and Mamba offers lower computational cost, making both architectures attractive depending on the resource constraints.
>
> Overall, our aim is not to determine which backbone universally dominates, but to show that GenKSR functions reliably with both architectures---Transformer offering higher expressivity, and Mamba offering greater computational efficiency. Extending these comparisons to larger qubit counts would require approximate simulators or larger-scale hardware experiments, which we view as a valuable direction for future work.

---

> ### Comment · Reviewer_c4Tv · 2025-11-25
>
> W1: New 2D experiments strengthen the empirical results and validate the generality of GenKSR.
>
> W2: Omission of ablation study is acceptable, but including these helps us understand the method better (why it works, what is the crucial component), which may inspire future research.
>
> W3: The original paper positions the choice of Mamba architecture as a major contribution. It seems to be still one of the main contributions (the second point) in the Introduction Section. But I think the existence of Transformer variants and the better performance of Transformer in 2D settings undermine this contribution.
>
> Q1: Thanks for the clarification.
>
> Q2: This aligns with my understanding, thanks.
>
> Q3: I wasn't asking about different **instances**, but rather **families**. It is made clear in the paper that it can generalize to unseen Hamiltonian instances in the same family. I am curious if it is possible to generalize across different Hamiltonian families, although I assume it would be difficult.
>
> Q4: Similar to W3. The original paper seems to emphasize a lot about the choice of Mamba.

---

> > ### Author Response · Authors · 2025-11-27
> > **Response to reviewer c4Tv**
> >
> > We thank Reviewer c4Tv for the prompt feedback and for acknowledging the value of our new 2D experiments. We address the remaining concerns below.
> >
> >
> >
> > **Response to W3 and Q4 (Positioning of Mamba)**
> >
> > We thank the reviewer for raising this point. We inadvertently left the second contribution point in the Introduction in its earlier Mamba-focused form. We appreciate the reviewer’s careful reading, which allowed us to correct this inconsistency.
> >
> > In the updated manuscript, the second contribution now emphasizes our comparative study of backbone architectures rather than positioning Mamba as the central architectural innovation. Together with revisions made in other parts of the manuscript, this clarifies that the core contribution of our work is GenKSR itself—the conditional generative modeling framework for quantum measurement distributions—which is inherently architecture-agnostic. Mamba is presented as one efficient option within this flexible framework, not as the exclusive centerpiece.
> >
> > We also note that in the original manuscript, we placed stronger emphasis on Mamba because it had not, to our knowledge, previously been explored in the context of generative modeling of quantum algorithms. Through the review process, it became clear that the architecture-agnostic nature of GenKSR is a more fundamental aspect of the contribution. Accordingly, we revised the manuscript to shift the emphasis.
> >
> >
> >
> > **Response to Q3 (Generalization across Hamiltonian Families)**
> >
> > We thank the reviewer for the clarification. We agree that generalization across different Hamiltonian families is an interesting but more challenging problem.
> >
> > In principle, GenKSR can support cross-family generalization to some extent, because the conditional network uses a graph-based Hamiltonian encoder that naturally processes two-body interaction graphs. This could enable the model to accommodate Hamiltonians that differ in geometry, coupling structure, and two-body Pauli interaction types. We revised Section 3.2 (Model Architecture) to clarify this point.
> >
> > However, we expect that achieving good generalization across physically disparate Hamiltonian families would require substantially broader training data and a conditional model with higher expressive capacity, and we view this as a natural direction in which GenKSR could be extended.
> >
> > For Hamiltonians whose interactions cannot be naturally represented using standard GCN-compatible graphs, GenKSR may require alternative encodings or more expressive neural architectures, such as attention-based models or message-passing networks. We regard these as possible extensions of the framework rather than limitations of the current contribution, and we have added a short discussion in the revised manuscript noting these as potential avenues for future work.
> >
> > We hope this addresses the reviewer's question.

---

> > > ### Comment · Reviewer_c4Tv · 2025-11-27
> > >
> > > Thank the authors for addressing my concerns. I now support acceptance for this paper. I have changed my scores accordingly.

---

### Author Response · Authors · 2025-11-23
**Summary of changes**

We would like to sincerely thank all Reviewers for their time and effort in evaluating our submission. The feedback has been very helpful in refining our presentation and expanding the scope of our experimental analysis.

We have carefully considered every point raised. In this revised manuscript, we have incorporated extensive new experimental results---most notably on 2D lattice systems---and significantly improved the discussions regarding baselines and architectural trade-offs. We believe these revisions firmly establish the generalization capabilities and practical utility of GenKSR.

Major changes in the revised manuscript:

**1. New Experiments on 2D Systems (Section 4.2 and Appendix A.4).**
To address concerns regarding the limitation to 1D models, we performed a new set of simulations on a 2D $J_1$-$J_2$ Heisenberg model on a $4\times4$ square lattice. We demonstrate that GenKSR successfully generalizes to unseen Hamiltonians in this complex 2D geometry and extrapolates to Krylov dimensions ($D=30$) twice as large as those seen during training ($D=15$).

**2. Comparison with Classical ML Solvers (ViT-NQS).**
Following the recommendation to compare against classical ML approaches, we added a direct comparison with a Vision Transformer-based Neural Quantum State (ViT-NQS) trained via Variational Monte Carlo. This case study (Section 4.2 and Appendix A.4) clarifies GenKSR's distinct role as a classical surrogate that faithfully mimics the quantum sampling process in KQD/SKQD, rather than a pure classical solver.

**3. Clarified Positioning of Mamba vs. Transformer.**
We have revised the Abstract, Introduction and conclusion to present Transformer and Mamba as two representative backbone architectures that illustrate a trade-off between accuracy (Transformer) and computational cost (Mamba). We removed claims suggesting Mamba strictly replaces Transformers, instead highlighting GenKSR as an architecture-agnostic framework.

**4. Correction on Time Embedding Mechanism.**
We corrected the description in Section 3.2 to explicitly state that the Krylov time step $t_l$ is mapped via a linear layer (treating it as a continuous scalar) rather than a lookup table. This clarifies the mechanism that enables our successful extrapolation to larger $D$.

We hope these revisions and detailed responses address your concerns. We are happy to engage in further discussion during the rebuttal period.

---

### Meta-Review · Area_Chair_m5kE · 2025-12-31

**Summary:**

The authors introduce Generative Krylov Subspace Representations (GenKSR), a framework that learns a classical generative representation of a Krylov diagonalization process used in estimating the energies of certain quantum states. While the use of generative AI in quantum information theory is interesting, the authors are completely open about the GenKSR consistently underperforming the AI-free "Classical Shadow" baseline in almost all scenarios (with shot counts of 5000 and higher, which is necessary for all but the smallest quantum systems). It would be most interesting to develop such methods further to present a clear advatage.

**Reviewer Concerns:**

In response to c4Tv, the authors have added substantial amount of 2D experiments and results into Section 4.2 and Appendix A.4 of the revised manuscript to support their claims of generalization across lattice geometries.

In response to DHK1 (W3), the authors discuss the benefits over AI-free baselines. The discussion is underwhelming in both the rebuttal and the revision.

All reviewers complained about the positioning of the use of the Mamba architecture to overcome the complexity of Transformer. This has not been addressed particularly well in either the rebuttal or the revision.

**Reviewer Scores:**

Two of the reviewers (DHK1, c4Tv) suggested that their scores would be increased. I doubt any of the reviewers would lower their scores.

---

### Decision · Program_Chairs · 2026-01-26

Reject